# QuanONet: Quantum Neural Operator with Application to Differential Equation

**Ruocheng Wang** [1]   **Zhuo Xia** [1]   **Ge Yan** [1]   **Junchi Yan** [1]

## Abstract

Differential equations are essential and popular in science and engineering. Learning-based methods including neural operators, have emerged as a promising paradigm. We explore its quantum counterpart, and propose QuanONet – a quantum neural operator which has not been well studied in literature compared with their counterparts in other machine learning areas. We design a novel architecture as a hardware-efficient ansatz, in the era of noisy intermediate-scale quantum (NISQ). Its circuit is pure quantum. By lying its ground on the operator approximation theorem for its quantum counterpart, QuanONet in theory can fit various differential equation operators. We also propose its modified version TF-QuanONet with ability to adaptively fit the dominant frequency of the problem. The real-device empirical results on problems including anti-derivative operators, Diffusion-reaction Systems demonstrate that QuanONet outperforms peer quantum methods when their model sizes are set akin to QuanONet.

## 1. Introduction

Nonlinear differential equations are essential tools in modeling complex phenomena across various scientific and engineering fields. Traditional numerical methods, including finite element methods (FEM) and finite difference methods (FDM), often rely on discretizing the problem domain and solving it iteratively. Recently, artificial intelligence (AI) methods, such as neural operators (Lu et al., 2021; Li et al., 2020), have emerged as promising alternatives by learning nonlinear operators directly and achieving some degree of generalization across different conditions.

Quantum computing offers a fundamentally different approach to addressing these challenges. Some existing quantum algorithms for solving differential equations map equations into higher or infinite dimensional linear spaces (Carleman, 1932; Forets & Pouly, 2017; Kowalski & Steeb, 1991; Jin et al., 2024) or use gradient computation of differentiable quantum circuits (Kyriienko et al., 2021; Xiao et al., 2024b) to solve differential equations. However, these methods are inherently limited to solving specific differential equations and lack the ability to learn nonlinear operators or generalize across varying conditions. This restricts their applicability in solving more complex real-world problems.

Quantum neural networks (QNNs) leverage the exponentially large Hilbert space of quantum systems to achieve quadratic speedup in online perceptron (Kapoor et al., 2016) and reinforcement learning (Dunjko et al., 2016). So there have been many studies using QNN to implement popular classical neural network architectures. However, existing QNN implementations of neural operators, such as QFNO (Jain et al., 2024) and quantum DeepONet (Xiao et al., 2024a), are limited to accelerating the evaluation phase using quantum Fourier transforms (QFT) and quantum orthogonal neural networks. The former is not observed to outperform classical operators on differential equation problems, and the latter uses a hybrid network structure that brings the communication overhead between quantum and classical devices.

We propose QuanONet, a general framework for learning (nonlinear) operators in differential equations using QNN. This means that it can not only be well-suited for the era of noisy intermediate-scale quantum (NISQ) by using any QNN architecture that features low qubit requirements and does not rely on high-connectivity quantum hardware, but also has potential in the future fault-tolerant quantum era.

Our QuanONet is grounded in the Universal Approximation Theorem for operators, and we show that its QNN implementation retains the powerful operator approximation capabilities of classical methods and have good generalization performance in different cases. Furthermore, we propose TF-QuanONet, a version of QuanONet that uses the trainable-frequency method (Jaderberg et al., 2024). By using the input encoded coefficients as additional trainable parameters, it can efficiently learn the spectrum of operators without the need for extremely deep quantum circuit repetition. These features position QuanONet as a versatile

[1]School of Artificial Intelligence & School of Computer Science & Zhiyuan College, SJTU. Correspondence to: Junchi Yan <yanjunchi@sjtu.edu.cn>.

*Proceedings of the $42^{nd}$ International Conference on Machine Learning*, Vancouver, Canada. PMLR 267, 2025. Copyright 2025 by the author(s).

and scalable tool for solving differential equations, with significant potential applications in fields such as fluid dynamics, material science, and physical simulation. **The contributions of this paper are:**

1) We extend the classic universal approximation theorem for operators (Chen & Chen, 1995) to the quantum state version, and give the universal approximation theorem for quantum computing for operators – see Theorem 3.1 and 3.2.

2) Based on our theoretical results, we propose a general trainable quantum neural network framework with pure quantum circuits without hybrid classic-quantum architecture as a nontrivial limitation suffered from existing work (Xiao et al., 2024a) namely QuanONet for learning (nonlinear) operators.

3) We further propose a trainable-frequency version, TF-QuanONet, which is tailored to dominant frequency component learning in operators and is different from all previous neural operators. It can capture the high frequency part of the operator without depending on the well setting of coefficients, and is thus robust and efficient.

4) We study a number of differential equation problems. Experimental results show that QuanONet with appropriate coefficient initialization can achieve better performance than other quantum solvers, while TF-QuanONet overcomes the difficulty of QuanONet in selecting appropriate hyperparameter coefficients, and performs well across various coefficient initial conditions.

## 2. Related Work and Preliminaries

### 2.1. Quantum Computing for Complex Problems

Quantum computing has shown great promise in solving complex problems, e.g. combinatorial optimization and quantum chemistry, whereby quantum variational algorithms like Quantum Approximate Optimization Algorithm (QAOA) (Farhi et al., 2014) have been well developed.

For differential equations solving, emerging quantum methods have been devised, which can be broadly classified into two categories. One is quantum algorithms based on amplitude methods and phase estimation, For example, Jin et al. (2024) converted linear differential equations with non-unitary dynamics into Schrödinger-type equations and solved using quantum phase estimation and amplitude amplification. Oz et al. (2023) used the quantum amplitude estimation by utilizing Chebyshev points for numerical integration to design robust quantum PDE solvers.

Another category is based on variable weight algorithm (VQA). Sarma et al. (2024); Lubasch et al. (2020) represented solution function with the state amplitudes and optimized the cost function, which is its difference between

the solution function obtained by the classical numerical method, to obtain the evolution operator of the equation in a short time. Some works (Kyriienko et al., 2021; Xiao et al., 2024b; Joo & Moon, 2021) utilized methods similar to Physics-Informed Neural Networks (PINNs) (Raissi et al., 2019). In this paradigm, quantum circuits are employed for variational optimization of differential equations, where quantum systems compute gradients through circuit evaluations. These gradients are then used to optimize neural network parameters, approximating solutions to the equations. While effective for specific problems, such quantum approaches have not yet been extended to the learning of nonlinear operators directly from data, an area that remains largely unexplored. More comprehensive discussion and comparison with related works are given in Appendix B.

Beyond differential equations, quantum computing has demonstrated potential in other complex problem domains. In combinatorial optimization, quantum algorithms like QAOA aim to find near-optimal solutions to problems such as graph partitioning (Herrman et al., 2021), scheduling and logistics (Azad et al., 2022). Similarly, in quantum chemistry, quantum algorithms are used to simulate molecular structures (Yan et al., 2023) and energy states (Cerezo et al., 2021) with greater precision than classical methods, potentially revolutionizing fields like drug discovery and materials science. These advancements highlight the versatility of quantum computing in addressing a wide range of computationally demanding problems.

### 2.2. Universal Approximation Theorems of QNNs

The foundation for the approximation capabilities of QNNs has been recently studied. Yu et al. (2022) showed that a QNN, even with a single qubit, can approximate any continuous single-variable function with arbitrary accuracy. This result parallels the Universal Approximation Theorem (Cybenko, 1989; Hornik, 1991) in classical neural nets. Moreover, Pérez-Salinas et al. (2020); Kapoor et al. (2016) proved a universal approximation theorem for QNNs by constructing a one-qubit quantum circuit able to arbitrarily approximate any continuous complex-valued function. Schuld et al. (2021) showed a similar approach that data encoding can be approximated by infinitely (akin to infinite-width classical neural networks) repeating simple encoding schemes as:

**Theorem 2.1.** *(Schuld et al., 2021) Denote $H_m$ a universal Hamiltonian family, $f_m$ the associated quantum model family:*

$$f_m(\mathbf{x}) = \langle \Gamma | S_{H_m}^\dagger(\mathbf{x}) M S_{H_m}(\mathbf{x}) | \Gamma \rangle \tag{1}$$

*For all functions $g \in L_2([0, 2\pi]^N)$, and for all $\epsilon > 0$, there exists some $m' \in \mathbb{N}$, some state $|\Gamma_i\rangle \in \mathbb{C}^{d^{m'}}$, and some Hamiltonian $M$ such that*

$$|f_{m'}(\mathbf{x}) - g(\mathbf{x})| \le \epsilon \tag{2}$$

Gonon & Jacquier (2023) moved one step further and proved precise error bounds for these approximations. Yu et al. (2024) showed that quantum models outperform classical ReLU neural networks by model size, circuit depth, and the number of trainable parameters when approximating high-dimensional functions with specific smoothness conditions. In addition, several works (Goto et al., 2021; Pérez-Salinas et al., 2021) have proposed that hybrid quantum classical architectures could provide a promising solution. By integrating trainable weights into the QNN, these models enrich the frequency spectrum of the network, enabling effective multivariable function approximation.

The frequency principle (Xu, 2018; Xu et al., 2019) suggested that DNNs train low-frequency components quickly but have poor generalization performance for high-frequency data. Its quantum version Xu & Zhang (2024) shows that QNNs preferentially learn the dominant frequency within the spectrum. In this paper, we study the training behavior in the specific operator learning context, especially for the high-frequency terms of operator.

### 2.3. The Deep Operator Network (DeepONet)

DeepONet is a deep learning framework designed to approximate nonlinear operators that map inputs (e.g., initial or boundary conditions) to the solutions of differential equations. Unlike traditional machine learning models that predict outputs for fixed input-output pairs, DeepONet learns the operator itself, mapping an input function to its corresponding output function. Its network structure is inspired by the universal approximation theorem for operator.

**Theorem 2.2.** *(Chen & Chen, 1995) Suppose that $g$ is a Tauber-Wiener function, $X$ is a Banach Space, $K_1 \subseteq X$, $K_2 \subseteq \mathbb{R}^n$ are two compact sets in $X$ and $\mathbb{R}^n$ respectively, $V$ is a compact set in $C(K_1)$, $G$ is a nonlinear continuous operator, which maps $V$ into $C(K_2)$. Then for any $\epsilon > 0$, there are positive integers $M, N, m$, constants $c_i^k, \xi_{ij}^k, \theta_i^k, \zeta_k \in \mathbb{R}$, $\boldsymbol{\omega}_k \in \mathbb{R}^n$, $\mathbf{x}_j \in K_1$, $i = 1, \cdots, M$, $k = 1, \cdots, N$, $j = 1, \cdots, m$, such that*

$$|G(u)(\mathbf{y}) - \sum_{k=1}^{N} \underbrace{\sum_{i=1}^{M} c_i^k g(\sum_{j=1}^{m} \xi_{ij}^k u(\mathbf{x}_j) + \theta_i^k)}_{Branch} \underbrace{g(\boldsymbol{\omega}_k \cdot \mathbf{y} + \zeta_k)}_{Trunk}| < \epsilon$$

(3)

As shown in Fig. 1, DeepONet consists of two primary components: Branch Net and Trunk Net. The Branch Net encodes the input function (e.g. initial conditions or boundary conditions) at a fixed number of sensors and the Trunk Net encodes the locations for the output functions. The output of the network is obtained by combining the outputs of the Branch and Trunk Nets, through a dot product, to generate the final solution of the operator. The model is trained on pairs of input-output data, generated from numerical solutions of differential equations.

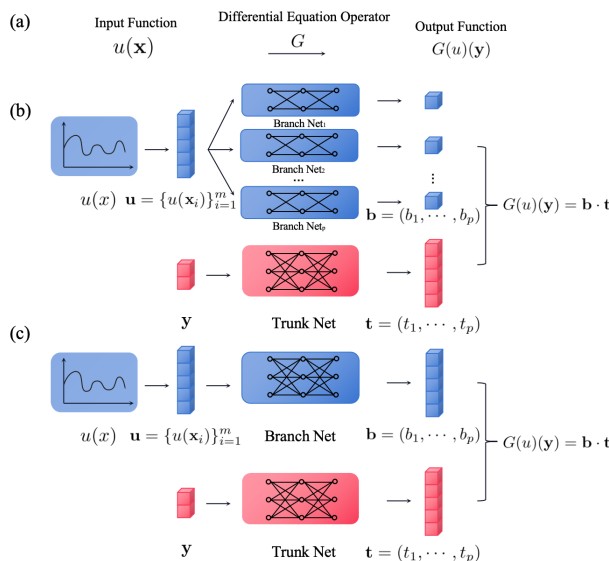

*Figure 1.* (a) For a differential equation, the solution operator $G$ tries to learn a mapping from function $u(x)$ to $G(u)(\mathbf{y})$. (b) Stacked DeepONet consists of a number of Branch Nets and a Trunk Net, following the form of Theorem 2.2. (c) Unstacked DeepONet is proposed by Lu et al. (2021) to reduce the storage and computation cost of Stacked DeepONet.

It has been shown that DeepONet can generalize to new, previously unseen conditions, enabling the rapid solution of differential equations with different initial or boundary conditions without retraining. At present, there have been many improvements for the original version, such as Lu et al. (2021) merging a large number of Branch Nets into one network to reduce memory and computational cost, called UnStacked DeepONet. Goswami et al. (2022) proposed a physical-informed variational formulation of DeepONet (V-DeepONet) to alleviate the computational burden. He et al. (2024) proposed the Sequential DeepONet (S-DeepONet) by introducing sequential learning models in the Branch Net to allow for accurate predictions of the solution contour plots under parametric and time-dependent loading histories.

## 3. Methodology

We will first derive the approximation theorem for quantum state functions, provide the quantum state substitution for the vector inner product in the classic operator approximation theorem and give the quantum universal approximation theorem for operators. The theory results inspire us to then propose a operator learning QNN architecture: QuanONet. We also discuss how the coding coefficients affect the approximation error of QuanONet through frequency spectral and how to overcome this effect.

## 3.1. Theoretical Results

### 3.1.1. QUANTUM UNIVERSAL APPROXIMATION FOR QUANTUM STATE FUNCTIONS

Although works about QNNs have studied universal approximation for functions, there still lack the approximation theorem for quantum state functions. Specifically, for a normalized quantum state that depends on the inputs, parametric quantum circuits can output its approximation with arbitrary accuracy, where the inputs are encoded as model parameters. Based on feasible assumptions about QNN (Schuld et al., 2021), we give an approximation scheme for an arbitrary quantum state function with the form of:

$$|\phi(\mathbf{x})\rangle = \sum_j g_j(\mathbf{x})|j\rangle \qquad (4)$$

It satisfies the normalization conditions $\sum_j |g_j(\mathbf{x}_j)|^2 = 1$. $\{|j\rangle\}$ is the basis of a finite dimensional quantum system.

Then we define a quantum state as the output of a parametric quantum circuit as follows:

$$|\psi_{\boldsymbol{\theta}}(\mathbf{x})\rangle = U(\boldsymbol{\theta}, \mathbf{x})|0\rangle \qquad (5)$$

where $\boldsymbol{\theta}$ and $\mathbf{x}$ are the optimizable parameters and the encoding parameters.

The quantum circuit consists of a data encoding circuit $S(\mathbf{x})$ and a trainable circuit $W(\boldsymbol{\theta})$ controlled by parameters $\boldsymbol{\theta}$ both alternating $L$ layers. The data encoding block consists of gates of the form $e^{-ixH}$. We focus on the role of the data encoding, and view the trainable circuit blocks as arbitrary unitary operations as $W(\boldsymbol{\theta}) = W$. Thus the overall quantum circuit has the form:

$$U(\mathbf{x}) = W^{(L+1)}S(\mathbf{x})W^{(L)}\cdots W^{(2)}S(\mathbf{x})W^{(1)} \qquad (6)$$

For notational simplicity and without loss of generality, we consider simple single-layer quantum circuits with $L = 1$, obtain $|\Gamma\rangle$ by acting $W^{(1)}$ on the initial state $|0\rangle$ and rewrite $W^{(2)}$ as $W$. Thus we have the equivalent form:

$$|\psi(\mathbf{x})\rangle = WS(\mathbf{x})|\Gamma\rangle \qquad (7)$$

Specifically, Schuld et al. (2021) introduced the concept of a Hamiltonian family $\{H_m|m \in \mathbb{N}\}$ where $H_m$ acts on $m$ subsystems of dimension $d$. Such a Hamiltonian family defines a family of quantum states via

$$|\psi_m(\mathbf{x})\rangle = WS_{H_m}(\mathbf{x})|\Gamma\rangle \qquad (8)$$

Given Hamiltonian $H_m$ with eigenvalues $\{\lambda_1, \cdots, \lambda_{d^m}\}$, the eigenspectrum of $H_m$ is

$$\Lambda_{H_m} = \{\lambda_j|j \in \{1, \cdots, d^m\}\} \qquad (9)$$

A Hamiltonian family whose eigenspectrum asymptotically contains any integer frequency via the following notion (Schuld et al., 2021): a Hamiltonian family $\{H_m\}$ is a universal Hamiltonian family if it has the property that for all $K \in \mathbb{N}$ there exists some $m \in \mathbb{N}$ such that

$$\mathbb{Z}_K = \{-K, \cdots, 0, \cdots, K\} \subseteq \Lambda_{H_m}. \qquad (10)$$

As proposed in Schuld et al. (2021), the Hamiltonian family defined as the linear combinations of Pauli operators

$$H_m = \sum_{i=1}^m \sigma_q^{(i)}, q \in \{x, y, z\}, \qquad (11)$$

is indeed a universal Hamiltonian family with $m = K$. This gives the theorem:

**Theorem 3.1.** *[Quantum Universal Approximation Theorem for State Functions]* Let $H_m$ be a universal Hamiltonian family, and $\psi_m$ the associated quantum state family:

$$|\psi_m(\mathbf{x})\rangle = WS_{H_m}(\mathbf{x})|\Gamma\rangle. \qquad (12)$$

*For all normalized quantum state functions $|\phi(\mathbf{x})\rangle = \sum_j g_j(\mathbf{x})|j\rangle$ with functions $g_j \in L_2([0, 2\pi]^N)$, and for all $\epsilon > 0$, there exists some $m' \in \mathbb{N}$, some state $|\Gamma\rangle \in \mathbb{C}^{d^{m'}}$, and some $W$ such that*

$$\||\psi_{m'}(\mathbf{x})\rangle - |\phi(\mathbf{x})\rangle\| \le \epsilon \qquad (13)$$

Compared with Theorem 2.1, the proof of Theorem 3.1 requires the spectrum $\Lambda_{H_m}$ be able to contain the highest integer frequency $K_s = \max K_j$ among all amplitudes in the state function. See proof in Appendix D.

Theorem 2.1 has a heuristic significance for the wider application of QNN: it shows that QNN cannot only be used for approximating multivariable functions, but also for learning quantum states that depend on the input, e.g. predicting ground state of quantum systems under different conditions. Based on this theorem, we will provide the construction of the quantum neural operator in the next section.

### 3.1.2. QUANTUM UNIVERSAL APPROXIMATION FOR OPERATORS

Based on the above theorem, we give the quantum universal approximation theorem for operators as follows.

**Theorem 3.2.** *[Quantum Universal Approximation Theorem for Operators]* Suppose that $X$ is a Banach Space, $K_1 \subseteq X$, $K_2 \subseteq \mathbb{R}^n$ are two compact sets in $X$ and $\mathbb{R}^n$ respectively, $V$ is a compact set in $C(K_1)$, $G$ is a nonlinear continuous operator, which maps $V$ into $C(K_2)$, then for any $\epsilon > 0$, there exist a positive integer $N$, a real constant $c$, a $N$-dim state function $|t\rangle$ and a $N$-dim state functional $|b\rangle$, such that

$$|G(u)(\mathbf{y}) - c\langle b(u)|t(\mathbf{y})\rangle| < \epsilon \qquad (14)$$

A full proof is given in Appendix E. This theorem states that the operator can be approximated by an inner product of quantum states depending on the input function and the output location respectively, implying the existence of a QNN structure which can learn the differential operator.

Before introducing the architecture of QuanONet inspired by the theorem, we briefly introduce and extend the error analysis of DeepONet with Lanthaler et al. (2022). Specifically, the error sources of DeepONet mainly include three parts: encoding, approximation and re-construction errors, where encoding error depends on the settings of sensors, However, both approximation and re-construction errors depend on the approximation ability of the network and the dimension of the vector (quantum state). Similarly, the error sources of QuanONet also come from these three parts, which inspires us that QuanONet may be applicable to the operator version of the QNN dominance problem, namely smooth and spectrally simple operators according to the results of Yu et al. (2024) and Leong et al. (2025).

### 3.2. The Architecture of QuanONet

As shown in Fig. 2, QuanONet consists of two primary components: Branch (BNC) Layers and Trunk (TNK) Layers, which encode the input function $\mathbf{u} = \{u(\mathbf{x}_i)\}_{i=1}^m$ and the output locations $\mathbf{y}$ respectively. The output of the circuit is:

$$\langle\psi_{\boldsymbol{\theta}}(\mathbf{y}, \mathbf{u})|H|\psi_{\boldsymbol{\theta}}(\mathbf{y}, \mathbf{u})\rangle$$
$$=\langle 0|U_{\text{Bnc}}^{\dagger}(\boldsymbol{\theta}^b, \mathbf{u})U_{\text{Tnk}}^{\dagger}(\boldsymbol{\theta}^t, \mathbf{y})HU_{\text{Tnk}}(\boldsymbol{\theta}^t, \mathbf{y})U_{\text{Bnc}}(\boldsymbol{\theta}^b, \mathbf{u})|0\rangle \tag{15}$$

Write Hamiltonian $H$ in spectral decomposition form $H = \sum_i \lambda_i |i\rangle\langle i|$, then the output can be written as

$$\langle\psi_{\boldsymbol{\theta}}(\mathbf{y}, \mathbf{u})|H|\psi_{\boldsymbol{\theta}}(\mathbf{y}, \mathbf{u})\rangle$$
$$= \sum_i \lambda_i |\langle i|U_{\text{Tnk}}(\boldsymbol{\theta}^t, \mathbf{y})U_{\text{Bnc}}(\boldsymbol{\theta}^b, \mathbf{u})|0\rangle|^2 \tag{16}$$

If Hamiltonian has only one eigenstate $|0\rangle$ with eigenvalue 1 and all other eigenvalues are zero, then the output corresponds to the squared version of the Eq. 3.2.

$$\langle\psi_{\boldsymbol{\theta}}(\mathbf{y}, \mathbf{u})|H|\psi_{\boldsymbol{\theta}}(\mathbf{y}, \mathbf{u})\rangle = |\langle 0|U_{\text{Tnk}}(\boldsymbol{\theta}^t, \mathbf{y})U_{\text{Bnc}}(\boldsymbol{\theta}^b, \mathbf{u})|0\rangle|^2$$
$$= |\langle t(\boldsymbol{\theta}^t, \mathbf{y})|b(\boldsymbol{\theta}^b, \mathbf{u})\rangle|^2 \tag{17}$$

Since the Hamiltonian is free to choose, we only need to scale it so that the bound of its spectrum includes the range of the output function. An optional Hamiltonian is:

$$H = a\sum_{i=1}^n \sigma_z^{(i)} + b \tag{18}$$

with the upper bound $b + na$ and the lower bound $b - na$. $n$ is the total number of qubits.

Since parameters of the rotation quantum gates are periodic in the range $[0, 2\pi]$, inputs $\mathbf{u}$ and $\mathbf{y}$ should be encoded in a

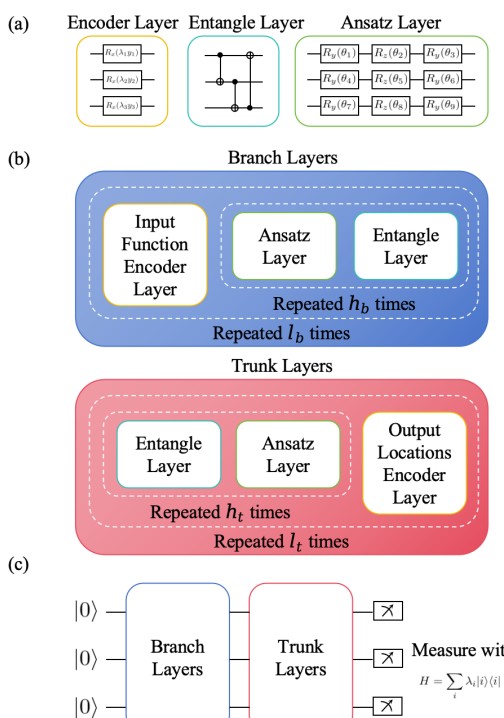

Figure 2. QuanONet's detailed architecture. (a) A hardware-efficient class of examples of Encoder, Entangle, and Ansatz layers. (b) Structure of Branch and Trunk Layers with hyper-parameters $h_b, h_t, l_b, l_t$ representing the number of repetitions of Ansatz-Entangle and encoder-train alternating layers in them. To satisfy the requirement of inner product form, the quantum Layers of Branch and Trunk Layers are arranged in reverse order. (c) The overall quantum circuit structure, and the circuit parameters are trained by the gradient of the loss function of the output value.

range less than $2\pi$. A simple method is multiplying inputs by hyper-parameter coefficients $\boldsymbol{\lambda} = [\lambda, \cdots, \lambda]$ so that

$$\lambda\|[\mathbf{y}, \mathbf{u}]\|_1 \leq \pi \tag{19}$$

Ideally, we could set $\lambda$ very small to ensure that the model fits a wider range of inputs. However as proposed in Xu & Zhang (2024); Jaderberg et al. (2024), the extremely small $\lambda$ means that a higher number of repeated layers is required to achieve Fourier series completeness. How to choose the most appropriate $\lambda$ under the condition that the size of the quantum circuit is limited requires careful consideration.

The order of applying Branch and Trunk Layers does not affect the approximation formula. We adopt the sequence of encoding the input $\mathbf{u}$ with the Branch Layers first, followed by the encoding of variable $\mathbf{y}$ with the Trunk Layers.

As a high-level framework, QuanONet allows for different implementations. To accommodate the current hardware

limitations of quantum computers, e.g. restricted connectivity and depth constraints, we employ a QNN design with Hardware-Efficient Ansatz (HEA) as shown in Fig. 2 as our vanilla version. By leveraging this hardware-efficient circuit structure, QuanONet achieves a practical balance between expressiveness and resource requirements, especially remaining deployable on existing quantum hardware.

### 3.3. The Trainable-Frequency Technique

The frequency principle provides a perspective for understanding the training behavior and generalization ability of deep neural networks. Since the data input is introduced into the quantum state through trigonometric functions, So Frequency plays a more critical role in training and optimization. Specifically, (Xu & Zhang, 2024) developed an evolution equation for gradients and residue dynamics in the frequency domain by introducing quantum neural tangent kernels (QNTK) and empirically showed that the dominant frequency of the data set is learned first by QNNs.

Though asymptotically large fixed-frequency (FF) QNN are universal function approximators (Schuld et al., 2021), in reality, finite-sized quantum computers will permit models with only a finite range of frequencies. So the approximation effect of quantum variational circuits is strongly related to whether the frequency spectrum can match the spectrum of problems. In Jaderberg et al. (2024), trainable-frequency (TF) models are introduced, which extend the conventional fixed-frequency parameterization of quantum circuits. In typical QNN models, the generator Hamiltonians used for data encoding determine a fixed set of equally spaced frequencies. TF models add trainable parameters to the generator, allowing the circuit to dynamically adjust the frequency spectrum it represents. It improves the ability to adapt to the spectral characteristics of the target function.

We propose TF-QuanONet, a tailored and improved version of QuanONet. TF-QuanONet employs a trainable-frequency model for its Branch and Trunk Layers. By introducing trainable weights and biases in the quantum encoding that is $\boldsymbol{\lambda} \odot [\mathbf{y}, \mathbf{u}] + \mathbf{b}$, it dynamically adjusts frequency representation during training. This allows the model to adapt to high-frequency or irregularly spaced spectral features of operators that may not be captured by FF-QNN.

## 4. Experiments

All the numerical simulations are performed on a machine with 190GB memory, one physical CPU with 32 cores AMD Ryzen Threadripper 3970X CPU, and 5 GPUs (NV GeForce RTX 3090). We implement a Python quantum simulator without noise to simulate QNNs Based on MindSpore (Xu et al., 2024) framework, and designing classical neural networks using DeepXde (Lu et al., 2020) framework.

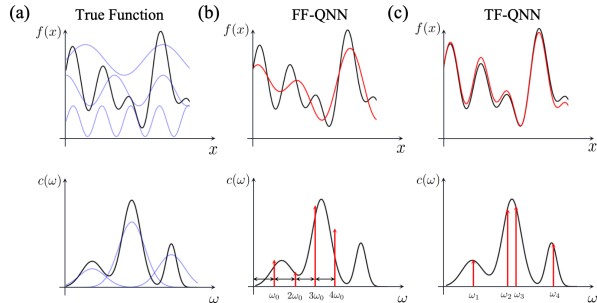

*Figure 3.* Schematic representation of the function and spectrum of the (a) true function, (b) FF-QNN and (c) TF-QNN, where the QNNs have been trained to the optimum. The FF-QNN model will have a large approximation error when it is not close to the complete Fourier series, while TF-QNN can preferentially fit the main frequency part to improve the accuracy. This effect is particularly significant when the high-frequency components of the function dominate or when the quantum circuit is shallow.

### 4.1. Protocols

We evaluate QuanONet and TF-QuanONet on a range of benchmark problems: three ordinary differential equation (ODE) and one partial differential equation (PDE) problem:

**Anti-derivative operator.**

$$G : u(x) \to v(x) = \int_0^x u(\tau) d\tau, x \in [0, 1] \quad (20)$$

**Homogeneous operator.**

$$G : u(x) \to v(x), \frac{dv(x)}{dx} = v(x) + u(x), x \in [0, 1] \quad (21)$$

**Nonlinear ODE.**

$$G : u(x) \to v(x), \frac{dv(x)}{dx} = -v(x)^2 + u(x), x \in [0, 1] \quad (22)$$

**Diffusion-reaction (D-R) System.**

$$G : u(x) \to v(x, t), \frac{\partial v}{\partial t} = D\frac{\partial^2 v}{\partial x^2} + kv^2 + u, x, t \in [0, 1] \quad (23)$$

all with zero initial/boundary conditions.

**Baseline.** We comprehensively compare the proposed QuanONet and TF-QuanONet with the following methods:

Quantum methods: QNN with hardware efficient ansatz (HEA) (Kandala et al., 2017), trainable-frequency HEA (TF-HEA) (Jaderberg et al., 2024).

Classic methods: fully-connected neural net (FNN) (Raissi et al., 2018)), DeepONet (Lu et al., 2021).

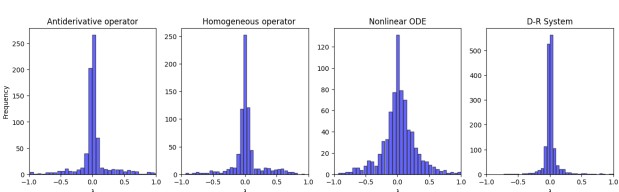

*Figure 4.* The $\lambda$ distribution of trained TF-QuanONet with initialized to 0.001 of all runs on each problem. The distribution in nonlinear ODE case is significantly more dispersed.

**Data Generation.** We generate the dataset by using the mean-zero Gaussian random field with the form: $u \sim \mathcal{G}(0, k_l(x_1, x_2))$ where the covariance kernel $k_l(x_1, x_2) = e^{-\|x_1 - x_2\|^2 / 2l^2}$ is the radial-basis function (RBF) kernel with a length-scale parameter $l > 0$ to sample input function $u$. We solve the ODE systems by Runge-Kutta (4, 5) and PDEs by a second-order finite difference method.

**Settings.** We scale the various QNNs to ensure that they have close number of parameters (1200 for ODE and 2400 for PDE) and use the Adam optimizer (Kingma, 2014) with learning rate 0.0001. The details are shown in the Appendix A. The number of qubits of all QNNs is 5, and the Hamiltonian is $H = \sum_{i=1}^{5} \sigma_z^{(i)}$. Detailed configuration is given in Table 2, Table 3 and Table 4.

### 4.2. Results

We use 5 runs with different training/test data sampling and network initialization to compute the mean error and the standard deviation (SD), as presented in Table 1.

For ODE e.g. homogeneous operator (Fig. 5), QuanONet performs best under $\lambda = 0.1$, but the error increases significantly as the hyperparameter $\lambda$ decreases. The trainable frequency technique improves TF-QuanONet's robustness to $\lambda$, and makes it perform optimally in all cases. Note that all quantum methods have less generalization error than FNN. In the case of nonlinear ODE, QNNs generally perform worse than classical methods, because this case has a more spread out frequency domain distribution compared to other problems (Fig. 4), so deeper QNNs are needed to ensure frequency completeness. But our method still enhances the best performance among quantum methods. Four representative cases are shown in Fig. 6 and Fig. 7.

The hardware noise characteristics in the NISQ era are primarily influenced by qubit count, gate cost and circuit depth. We conduct extensive benchmarking across various qubit count, layer depths and Hamiltonian, with complete experimental results and analysis presented in Appendix F.

Fig. 8 further supports the potential of TF-QuanONet on

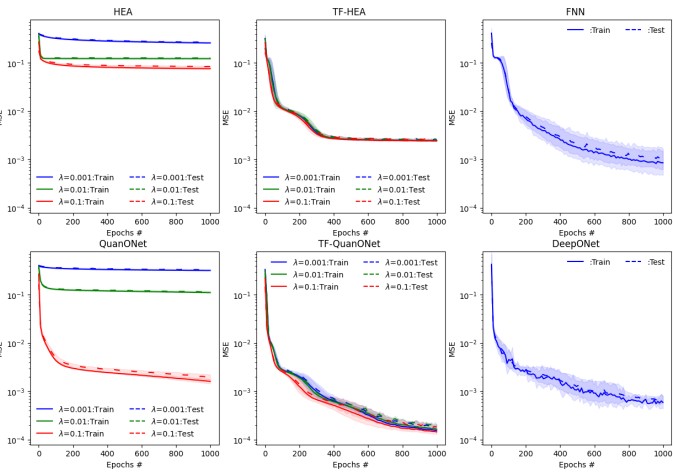

*Figure 5.* Errors of all methods trained to learn the homogeneous operator. The solid and dash lines are the training error and test error in training, respectively. The shaded regions are the one-standard-derivation from 5 runs with different training/test data and network initialization. All methods are plotted by a uniform ordinate scale.

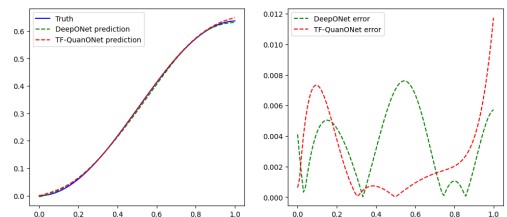

| Method | Branch | | Trunk | | Params. # |
|---|---|---|---|---|---|
| TF-QuanONet | Depth | Ansatz Depth | Depth | Ansatz Depth | 1200 |
| | 20 | 2 | 10 | 2 | |
| DeepONet | Depth | Width | Depth | Width | 1261 |
| | 2 | 10 | 2 | 10 | |

*Figure 6.* Anti-derivative operator results for the input function $u = \sin \pi x$, with hyperparameters of networks in below.

real-devices. We train a 2-qubits TF-QuanONet for the anti-derivative operator and test it on IBM brisbane Q57/Q58[1]. By leveraging Qiskit's compilation optimization techniques and using standard noise mitigation, the circuit depth is reduced to merely 20 layers. Using $u(x) = x$ as input function and standard noise mitigation like zero-noise extrapolation (Giurgica-Tiron et al., 2020), we achieved MSE = 1.57e-3 (vs. 1.5e-5 simulation). The gap is mainly attributed to non-ideal gate operations and residual decoherence effects.

---

[1]We acknowledge Shikun Wei who helped the IBM Quantum platform access and experiments during his stay in Paris, France.

*Table 1.* Test error comparison with different initial values of $\lambda$. The best is in gray and second best in lightgray.

| Method | $\lambda$ | ODE | | | PDE |
|---|---|---|---|---|---|
| | | Antideriv operator | Homogeneous operator | Nonlinear ODE | D-R System |
| HEA | 0.001 | $0.095871 \pm 0.002140$ | $0.267378 \pm 0.006169$ | $0.900696 \pm 0.034579$ | $0.000601 \pm 0.000022$ |
| | 0.01 | $0.053324 \pm 0.001074$ | $0.127268 \pm 0.001941$ | $0.705380 \pm 0.030721$ | $0.000530 \pm 0.000039$ |
| | 0.1 | $0.030112 \pm 0.000759$ | $0.084640 \pm 0.002567$ | $0.369845 \pm 0.017249$ | $0.006988 \pm 0.001866$ |
| TF-HEA | 0.001 | $0.002484 \pm 0.000028$ | $0.002608 \pm 0.000048$ | $0.045553 \pm 0.007749$ | $0.000113 \pm 0.000021$ |
| | 0.01 | $0.002518 \pm 0.000040$ | $0.002604 \pm 0.000039$ | $0.044619 \pm 0.008089$ | $0.000102 \pm 0.000008$ |
| | 0.1 | $0.002519 \pm 0.000035$ | $0.002613 \pm 0.000032$ | $0.044673 \pm 0.007877$ | $0.000807 \pm 0.000131$ |
| QuanONet | 0.001 | $0.131876 \pm 0.003809$ | $0.330244 \pm 0.005671$ | $0.869680 \pm 0.031546$ | $0.000601 \pm 0.000033$ |
| | 0.01 | $0.043198 \pm 0.001002$ | $0.114915 \pm 0.003816$ | $0.630765 \pm 0.031184$ | $0.000347 \pm 0.000015$ |
| | 0.1 | $0.001341 \pm 0.000190$ | $0.002005 \pm 0.000253$ | $0.059582 \pm 0.008323$ | $0.001802 \pm 0.001053$ |
| TF-QuanONet | 0.001 | $0.000113 \pm 0.000010$ | $0.000192 \pm 0.000028$ | $0.039072 \pm 0.007277$ | $0.000055 \pm 0.000014$ |
| | 0.01 | $0.000121 \pm 0.000010$ | $0.000198 \pm 0.000013$ | $0.039469 \pm 0.008081$ | $0.000051 \pm 0.000010$ |
| | 0.1 | $0.000112 \pm 0.000008$ | $0.000181 \pm 0.000021$ | $0.039074 \pm 0.007452$ | $0.000145 \pm 0.000028$ |
| DeepONet | | $0.000596 \pm 0.000215$ | $0.000665 \pm 0.000117$ | $0.027237 \pm 0.003972$ | $0.000088 \pm 0.000036$ |
| FNN | | $0.000934 \pm 0.000594$ | $0.001187 \pm 0.000732$ | $0.028487 \pm 0.006250$ | $0.000089 \pm 0.000066$ |

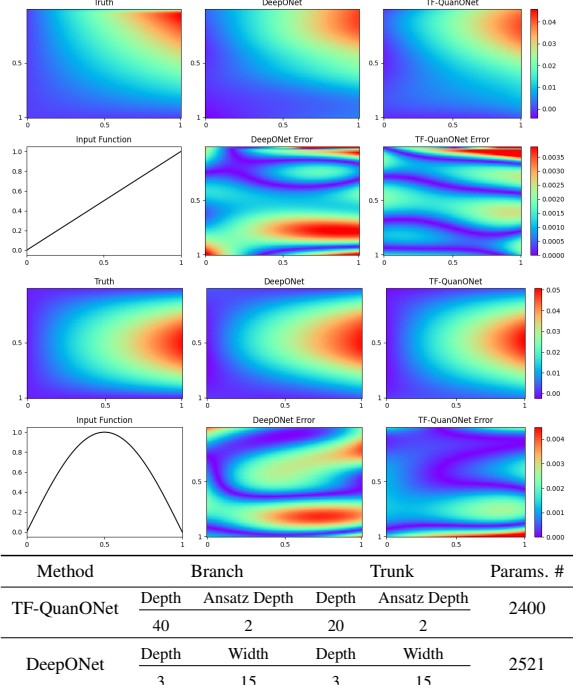

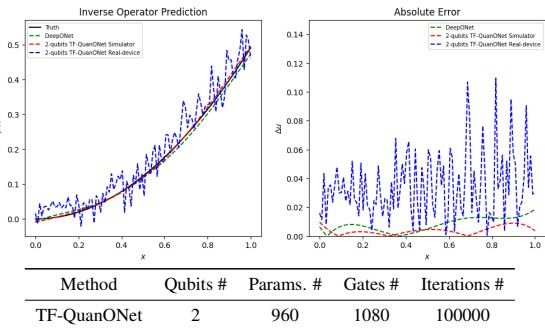

| Method | Qubits # | Params. # | Gates # | Iterations # |
|---|---|---|---|---|
| TF-QuanONet | 2 | 960 | 1080 | 100000 |

*Figure 8.* Inference results of a 2-qubits TF-QuanONet and DeepONet on anti-derivative operator. The input function is $u(x) = x$.

niques help TF-QuanONet learn the dominant frequency of the operator, thus providing robustness.

**Broad Applicability.** TF-QuanONet effectively handles various operator learning tasks, showcasing its versatility and potential for solving complex differential equations.

## 5. Conclusion and Outlook

We have derived the universal approximation theorem of QNNs for quantum state functions, giving a quantum version of the operator approximation theorem and constructing a quantum neural operator QuanONet based entirely on quantum circuits. To address the frequency spectral locality issue that may arise due to the coding coefficient setting, we introduce the trainable frequency method and propose an improved version: TF-QuanONet.

## Acknowledgment

Work was partly supported by NSFC (92370201, 62222607).

| Method | Branch | | Trunk | | Params. # |
|---|---|---|---|---|---|
| TF-QuanONet | Depth | Ansatz Depth | Depth | Ansatz Depth | 2400 |
| | 40 | 2 | 20 | 2 | |
| DeepONet | Depth | Width | Depth | Width | 2521 |
| | 3 | 15 | 3 | 15 | |

*Figure 7.* D-R system results for input functions $u = x$ (above) and $u = \sin \pi x$ (below). The hyperparameters of networks are shown below. The predictions and the truth share the same colorbar, and errors of two methods are plotted on another same colorbar.

The experiments collectively show that:

**Expressiveness.** TF-QuanONet achieves lower approximation errors over diverse tasks, outperforming existing quantum methods at similar scale and model size.

**Optimization Efficiency.** Coefficient setting significantly influences approximation error. Trainable frequency tech-

## Impact Statement

This paper concerns the between of machine learning and quantum computing for the fundamental problem of differential equation solving. It has wide technical impact to AI and other disciplines and we need to be careful the fast development of the potential technology.

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

# A. Details of the Experiments

Table 2. Default parameters for each problem

| | Case | $u$ space | Sensors # | Training # | Testing # | Batch Size |
|---|---|---|---|---|---|---|
| | Anti-deriv operator | GRF($l = 0.2$) | 100 | 10000 | 100000 | 100 |
| ODE | Homogeneous operator | GRF($l = 0.2$) | 100 | 10000 | 100000 | 100 |
| | Nonlinear ODE | GRF($l = 0.2$) | 100 | 10000 | 100000 | 100 |
| PDE | D-R system | GRF($l = 0.2$) | 100 | 100000 | 1000000 | 100 |

Table 3. Hyperparameters for each network in ODE case

| Network Type | Branch | | Trunk | | Gates # | Param.# | Iterations # |
|---|---|---|---|---|---|---|---|
| | Depth | Ansatz Depth | Depth | Ansatz Depth | | | |
| QuanONet | 20 | 2 | 20 | 2 | 1800 | 1200 | 100000 |
| TF-QuanONet | 20 | 2 | 10 | 2 | 1350 | 1200 | 100000 |
| | Depth | | Ansatz Depth | | | | |
| HEA | 40 | | 2 | | 1800 | 1200 | 100000 |
| TF-HEA | 32 | | 2 | | 1440 | 1280 | 100000 |
| | Branch | | Trunk | | | | |
| | Depth | Width | Depth | Width | | | |
| DeepONet | 2 | 10 | 2 | 10 | — | 1261 | 100000 |
| | Depth | | Width | | | | |
| FNN | 2 | | 10 | | — | 1251 | 100000 |

Table 4. Hyperparameters for each network in PDE case

| Network Type | Branch | | Trunk | | Gates # | Param.# | Iterations # |
|---|---|---|---|---|---|---|---|
| | Depth | Ansatz Depth | Depth | Ansatz Depth | | | |
| QuanONet | 40 | 2 | 40 | 2 | 3600 | 2400 | 100000 |
| TF-QuanONet | 40 | 2 | 20 | 2 | 2700 | 2400 | 100000 |
| | Depth | | Ansatz Depth | | | | |
| HEA | 80 | | 2 | | 2880 | 2400 | 100000 |
| TF-HEA | 64 | | 2 | | 3600 | 2560 | 100000 |
| | Branch | | Trunk | | | | |
| | Depth | Width | Depth | Width | | | |
| DeepONet | 3 | 15 | 3 | 15 | — | 2521 | 100000 |
| | Depth | | Width | | | | |
| FNN | 3 | | 16 | | — | 2481 | 100000 |

# B. Further Related Works

## B.1. Neural operators accelerated by quantum computing

DeepONet entails quadratic complexity concerning input dimensions during evaluation. Given the progress in quantum algorithms and hardware, Xiao et al. (2024a) proposed to utilize quantum computing to accelerate DeepONet evaluations, yielding complexity that is linear in input dimensions. The quantum DeepONet they proposed integrates unary encoding and orthogonal quantum layers. However, since the quantum DeepONet only utilizes quantum circuits to speed up DeepONet's

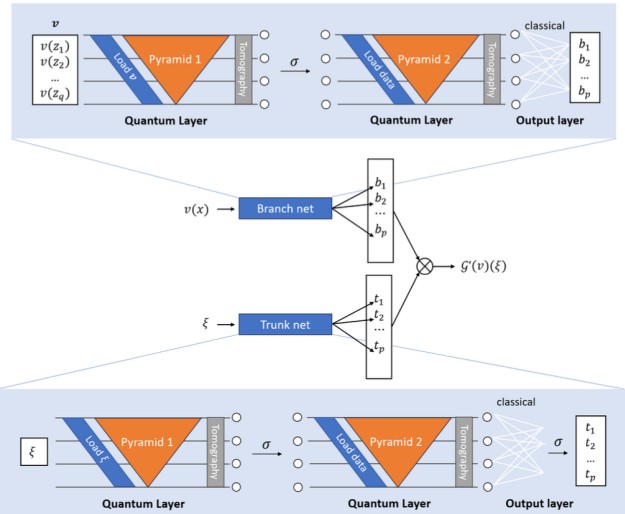

*Figure 9.* (**Xiao et al., 2024a**) Architecture of quantum DeepONet. The Branch Net and Trunk Net are replaced these with QOrthoNN, which is composed of several quantum layers arranged sequentially. The nonlinear operations are performed on classical computers.

evaluation, this means that we cannot expect it to achieve better results than DeepONet (or even worse if noise is taken into account), so we provide results of DeepONet as an alternative.

### B.2. Fourier neural operator with quantum Fourier layers

Jain et al. (2024) proposed a new quantum circuit for performing a quantum fourier transform (QFT) on an input encoded as a superposition of unary states which can be sequentially combined to form QFNO, a quantum version of a trainable Fourier neural network for solving Parametric PDEs.

(a)                    (b)

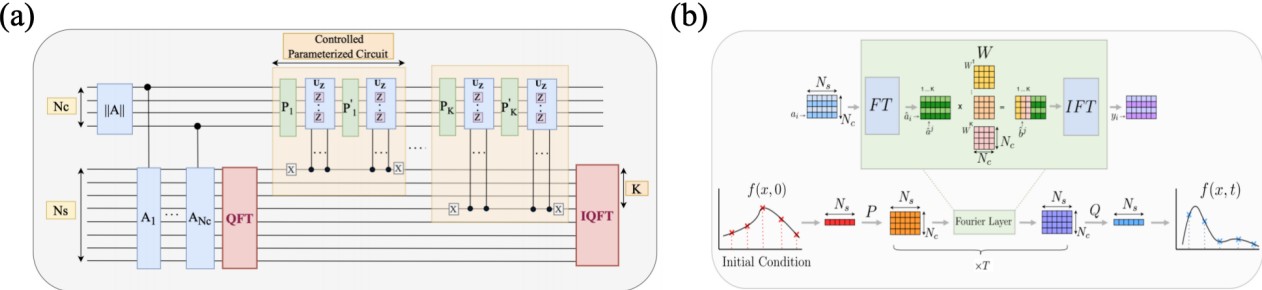

*Figure 10.* (**Jain et al., 2024**) (a) The proposed Sequential Quantum Circuit, which replicates the classical FNO operation from (b) if we measure at the end.

Compared with classical FNO, QFNO uses QFT to construct Fourier transform layer, but the advantage is not obvious. In numerical experiments, QFNO does not observe better performance than classical FNO. Jain et al. (2024) proposed that due to the efficiency of the QFT, QFNO has logarithmic time complexity compared to $N_s$ input dimension, while its classical counterpart has linear time complexity as shown in Table 5. However, the number of qubits and circuit depth of QFNO will still bring huge noise to current quantum devices, making the QFT impossible.

We provide the results of FNO with 100 initial functions as training instances, as shown in Tab. 6

*Table 5.* (**Jain et al., 2024**) Comparison of the order of time/depth complexities of the quantum Fourier Layer (QFL) in QFNO circuits with the classical Fourier Layer (FL). Here $N_s$ denotes the sampling dimension, $N_c$ denotes the channel dimension and $K$ denotes the maximum number of modes allowed. **If QFNO is used to solve the benchmark problem in this paper, it means that multiple low-noise quantum circuits with up to 100 qubits are needed, which cannot be realized using the existing quantum devices.**

| Method | Qubits # | Circuits # | Gate Complexity | Depth Complexity |
|---|---|---|---|---|
| Classical FL | — | — | — | $N_c + N_s \log N_s$ |
| Parallel QFL | $N_c + N_s$ | $K$ | $K N_c \log N_c + K N_c N_s \log N_s$ | $N_c + N_c \log N_s$ |
| Sequential QFL | $N_c + N_s$ | 1 | $K N_c \log N_c + N_c N_s \log N_s$ | $K N_c + N_c \log N_s$ |
| Composite QFL | $N_c + N_s$ | 1 | $(N_c + K) \log(N_c + K) + N_c N_s \log N_s$ | $\log(N_c + K) + N_c \log N_s$ |

*Table 6.* Test error comparison of TF-QuanONet, DeepONet and FNO

| Method | Data type | Iteration # | Antideriv operator | Homogeneous operator | Nonlinear ODE |
|---|---|---|---|---|---|
| TF-QuanONet | Aligned | 10000 | $0.002763 \pm 0.000107$ | $0.003424 \pm 0.000270$ | $0.114749 \pm 0.021397$ |
| | Semi-aligned | 10000 | $0.002524 \pm 0.000153$ | $0.003020 \pm 0.000151$ | $0.077709 \pm 0.011518$ |
| DeepONet | Aligned | 10000 | $0.004308 \pm 0.000332$ | $0.006661 \pm 0.002313$ | $0.856181 \pm 0.127804$ |
| | Semi-aligned | 10000 | $0.003557 \pm 0.000342$ | $0.006016 \pm 0.003654$ | $0.279827 \pm 0.071935$ |

| Method | Batch Size | Iteration # | Antideriv operator | Homogeneous operator | Nonlinear ODE |
|---|---|---|---|---|---|
| | 1 | 100000 | $0.031340 \pm 0.008230$ | $0.086708 \pm 0.029126$ | $0.330201 \pm 0.149577$ |
| FNO | 10 | 10000 | $0.004615 \pm 0.001619$ | $0.009267 \pm 0.002576$ | $0.077072 \pm 0.015786$ |
| | 50 | 2000 | $0.006853 \pm 0.001724$ | $0.015811 \pm 0.003861$ | $0.096177 \pm 0.026694$ |

## B.3. Physics-informed quantum neural network

For the output of a QNN as follows:

$$f_{\boldsymbol{\theta}}(\mathbf{x}) = \langle \psi_{\boldsymbol{\theta}}(\mathbf{x})|H|\psi_{\boldsymbol{\theta}}(\mathbf{x})\rangle \tag{24}$$

Define another two sets of inputs $\mathbf{x}^{(i)+}$ and $\mathbf{x}^{(i)-}$ as:

$$\mathbf{x}^{(i)+} = [x_0, x_1, \cdots, x_i + \frac{\pi}{2}, \cdots], \mathbf{x}^{(i)-} = [x_0, x_1, \cdots, x_i - \frac{\pi}{2}, \cdots] \tag{25}$$

Thus the partial derivative of $f(\mathbf{x})$ can easily calculated by

$$\nabla_{x_i} f_{\boldsymbol{\theta}}(\mathbf{x}) = \frac{1}{2}[f_{\boldsymbol{\theta}}(\mathbf{x}^{(i)+}) - f_{\boldsymbol{\theta}}(\mathbf{x}^{(i)-})] \tag{26}$$

Based on the differentiability of QNNs like this, Xiao et al. (2024b) introduced a physics-informed quantum neural network (PI-QNN) by employing the QNN as the function approximator for solving forward and inverse problems of PDEs. Numerical results indicate that PI-QNN demonstrates superior convergence over PINN when solving PDEs with exact solutions that are strongly correlated with trigonometric functions. Kyriienko et al. (2021) proposed a similar approach earlier and called it differentiable quantum circuits (DQC), and designed a Chebyshev quantum feature map that offers a powerful basis set of fitting polynomials and furnishings rich expressivity.

They are all quantum methods designed for solving differential equations with specific input functions, thus are fundamentally different from the quantum neural operators proposed in this paper.

## B.4. Fault-tolerant quantum algorithms for differential equations

Berry (2014) proposed a fault-tolerant quantum algorithm which solves sparse systems of linear ODEs by discretising the system of ODEs and subsequently employing the HHL algorithm (Harrow et al., 2009) to solve the resulting system of linear equations. Childs & Liu (2020) also proposed a quantum algorithm for solving linear ODEs, which, however, relies

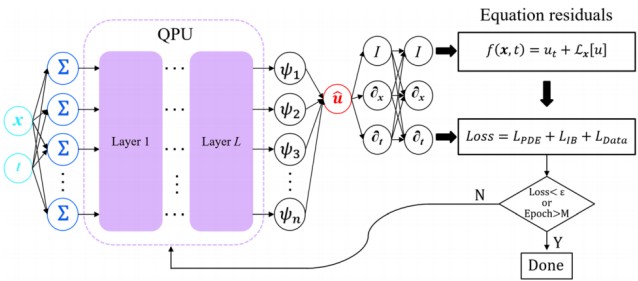

*Figure 11.* Schematic diagram of PI-QNN (Xiao et al., 2024b).

on spectral methods. Spectral methods use linear combinations of basis functions (e.g., a Fourier basis) to approximate the solution. This approach also ends with solving a linear system of equations on a quantum computer. A quantum algorithm to solve quadratically nonlinear ODEs under certain conditions is described in (Liu et al., 2021), using the Carleman linearisation (Carleman, 1932; Forets & Pouly, 2017; Kowalski & Steeb, 1991) to approximate the nonlinear part. The Carleman linearisation represents a finite-dimensional polynomially nonlinear system by an infinite-dimensional linear system. To make use of the Carleman linearisation, the infinite-dimensional linear system is truncated at a certain point. Subsequently, Liu et al. (2021) again discretised the resulting system and solve the linear system with HHL (Harrow et al., 2009). The algorithm presented in (Liu et al., 2021) may also be applied to solving certain PDEs (with a restricted kind of nonlinearity), as the discretisation of a PDE in all but one dimension generally results in a system of ODEs. Liu et al. (2023) presented an algorithm for solving (nonlinear) reaction-diffusion equations. Using Euler's method, as well as the Carleman linearisation for the nonlinearity, they discretise the PDE and solve the resulting system with HHL. Childs et al. (2021) presented quantum algorithms for solving linear PDEs by making use of the finite difference method (FDM) and spectral methods separately. In the FDM approach, they discretise the PDE on a grid. Both cases result in a linear system of equations which needs to be solved. The authors of (Arrazola et al., 2019) also present a quantum algorithm for solving linear PDEs, which relies on Hamiltonian simulation of a cleverly chosen Hamiltonian, which encodes certain properties of the PDE. Lloyd et al. (2020) outlined a quantum algorithm to solve nonlinear ODEs by mapping the ODE to the nonlinear Schrödinger equation, which is then solved using Trotterisation. Another numerical scheme, the finite element method (FEM, which approximates the solution by using interpolating functions within each discretised element) and HHL are used to solve elliptic PDEs in (Montanaro & Pallister, 2016). Jin & Liu (2022); Jin et al. (2023; 2024) derived quantum algorithms for solving nonlinear ODEs as well as the nonlinear Hamilton-Jacobi (HJ) equation (which is a special case of the nonlinear Hamilton-Jacobi-Bellman PDE). They do so, by mapping the nonlinear ODEs and the nonlinear HJ equation to linear ODEs and HJ equations using linear representation methods, such as the level set method, and then use HHL to solve the linear system.

## C. Partial Fourier Series Representation of State Functions

We show how a certain class of $L = 1$ quantum models naturally realise states with amplitudes of multivariate Fourier series.

We consider a output quantum state of the form:

$$|\psi(\mathbf{x})\rangle = WS(\mathbf{x})|\Gamma\rangle \tag{27}$$

with

$$S(\mathbf{x}) := e^{-ix_1 H_1} \otimes \cdots \otimes e^{-ix_N H_N} \tag{28}$$

where

$$|\Gamma\rangle = \sum_{j_1,\cdots,j_N=1}^{2^d} \gamma_{j_1,\cdots,j_N}|j_1\rangle \otimes \cdots \otimes |j_N\rangle \tag{29}$$

is some arbitrary state, and $W$ is some arbitrary unitary operator. To simplify the index handling, we introduce the

multi-indices $\mathbf{j} \in [2^d]^N$ with which we can rewrite

$$|\Gamma\rangle := \sum_{\mathbf{j}} \gamma_{\mathbf{j}} |\mathbf{j}\rangle \tag{30}$$

Additionally, as argued before we can without loss of generality assume that all Hamiltonians are diagonal, i.e.:

$$H_k = \mathrm{diag}(\lambda_1^{(k)}, \cdots, \lambda_{2^d}^{(k)}) \tag{31}$$

With this assumption, we note that $S(\mathbf{x})$ is diagonal with entries

$$[S(\mathbf{x})]_{\mathbf{j},\mathbf{j}} = e^{-i\mathbf{x}\cdot\boldsymbol{\lambda}_{\mathbf{j}}} \tag{32}$$

where we have defined

$$\boldsymbol{\lambda}_{\mathbf{j}} = (\lambda_{j_1}^{(1)}, \cdots, \lambda_{j_N}^{(N)}) \tag{33}$$

Given this, we see that

$$
\begin{aligned}
|\psi(\mathbf{x})\rangle &= \sum_{\mathbf{j}} \sum_{\mathbf{k}} \gamma_{\mathbf{j}} [WS(\mathbf{x})]_{\mathbf{j},\mathbf{k}} |\mathbf{j}\rangle \\
&= \sum_{\mathbf{j}} \sum_{\mathbf{k}} \gamma_{\mathbf{j}} W_{\mathbf{j},\mathbf{k}} e^{i\mathbf{x}\cdot\boldsymbol{\lambda}_{\mathbf{j}}} |\mathbf{j}\rangle
\end{aligned}
\tag{34}
$$

which is indeed a partial multivariate Fourier series, with the accessible frequencies fully determined by the spectra of the encoding Hamiltonians $\{H_k\}$, and the Fourier coefficients determined by the trainable unitaries (or equivalently, the state and Hamiltonian).

## D. Proof of the Quantum Universal Approximation Theorem for State Functions

We provide in this section a proof of the Theorem 3.1.

**Theorem 3.1** *Let $H_m$ be a universal Hamiltonian family, and $\psi_m$ the associated quantum state family:*

$$|\psi_m(\mathbf{x})\rangle = WS_{H_m}(\mathbf{x})|\Gamma\rangle. \tag{35}$$

*For all normalized quantum state functions $|\phi(\mathbf{x})\rangle = \sum_j g_j(\mathbf{x})|j\rangle$ with functions $g_j \in L_2([0, 2\pi]^N)$, and for all $\epsilon > 0$, there exists some $m' \in \mathbb{N}$, some state $|\Gamma\rangle \in \mathbb{C}^{d^{m'}}$, and some $W$ such that*

$$\||\psi_{m'}(\mathbf{x})\rangle - |\phi(\mathbf{x})\rangle\| \leq \epsilon \tag{36}$$

*Proof.* To begin with, we note that we can approximate any given $g \in L_2([0, 2\pi]^N)$, up to an arbitrarily small error in $L_2$ norm, by using a truncated Fourier series (Weisz, 2012). More specifically, for any $\epsilon > 0$, there exists some $K \in \mathbb{N}$ and some set of coefficients $\{c_{\mathbf{n}} | \mathbf{n} \in \mathbb{Z}_K^N\}$, such that

$$\tilde{g}(\mathbf{x}) = \sum_{n_1=-K}^{K} \cdots \sum_{n_N=-K}^{K} c_{\mathbf{n}} e^{i\mathbf{x}\cdot\mathbf{n}} := \sum_{\mathbf{n}\in\mathbb{Z}_K^N} c_{\mathbf{n}} e^{i\mathbf{x}\cdot\mathbf{n}} \tag{37}$$

satisfies

$$\|\tilde{g}(\mathbf{x}) - g(\mathbf{x})\| \leq \epsilon \tag{38}$$

Thus we rewrite $|\phi(\mathbf{x})\rangle$ with $\tilde{g}_j(\mathbf{x}) = \sum_{\mathbf{n}\in\mathbb{Z}_{K_j}^N} c_{j\mathbf{n}} e^{i\mathbf{x}\cdot\mathbf{n}}$

$$|\phi(\mathbf{x})\rangle = \sum_j \sum_{\mathbf{n}\in\mathbb{Z}_{K_j}^N} c_{j\mathbf{n}} e^{i\mathbf{x}\cdot\mathbf{n}} |j\rangle = \sum_{\mathbf{n}\in\mathbb{Z}_{K_s}^N} \sum_j c_{j\mathbf{n}} e^{i\mathbf{x}\cdot\mathbf{n}} |j\rangle \tag{39}$$

where $K_s = \max\{K_j\}$.

To prove the theorem we therefore only need to show that there exists an $m' \in \mathbb{N}$, some state $|\Gamma\rangle$ and some unitary operator $W$ so that the output quantum state $|\psi_m(\mathbf{x})\rangle$ generates the state function $|\phi(\mathbf{x})\rangle$. Recall that the output quantum state is defined as

$$|\psi_m(\mathbf{x})\rangle = WS_{H_m}(\mathbf{x})|\Gamma\rangle \tag{40}$$

with

$$S_{H_m}(\mathbf{x}) := e^{-ix_1 H_m} \otimes \cdots \otimes e^{-ix_N H_m} \tag{41}$$

In Appendix C , we have seen that we can express the output quantum state as

$$
\begin{aligned}
|\psi(\mathbf{x})\rangle &= \sum_{\mathbf{j}} \sum_{\mathbf{k}} \gamma_{\mathbf{j}} [WS(\mathbf{x})]_{\mathbf{j},\mathbf{k}} |\mathbf{j}\rangle \\
&= \sum_{\mathbf{j}} \sum_{\mathbf{k}} \gamma_{\mathbf{j}} W_{\mathbf{j},\mathbf{k}} e^{i\mathbf{x}\cdot\boldsymbol{\lambda}_{\mathbf{j}}} |\mathbf{j}\rangle
\end{aligned}
\tag{42}
$$

where the multi-indices $\mathbf{j}$ and $\mathbf{k}$ have $N$ entries that iterate over all $2^d$ basis states of the $d$ qubit subsystems. Let $\Lambda_{H_m}$ be the eigenspectrum of $H_m$, as defined in Eq. 9. As the $\{H_m\}$ form a universal family of Hamiltonians by assumption, we can choose an $m' \in \mathbb{N}$ so that

$$\mathbb{Z}_{K_s} = \{-K_s, \cdots, 0 \cdots, K_s\} \subseteq \Lambda_{H'_m} \tag{43}$$

The accessible frequency vectors $\boldsymbol{\lambda}_{\mathbf{j}}$ independently contain all possible combinations of the frequencies in $\Lambda_{H'_m}$.

The vector-valued frequency spectrum for the multivariate case is therefore the Cartesian product of $N$ copies of $\Lambda_{H'_m}$:

$$\Omega = \underbrace{\Lambda_{H_{m'}} \times \cdots \times \Lambda_{H_{m'}}}_{N \text{ times}} \tag{44}$$

As $\mathbb{Z}_K \subseteq \Lambda_{H'_m}$ we naturally have that $\mathbb{Z}_{K_s}^N \subseteq \Omega$, which means that the Fourier series generated by the chosen model contains all terms that are necessary to construct the Fourier series state $|\phi(\mathbf{x})\rangle$. With the coefficients which can be freely chosen, Eq. 42 can be easily chosen to be the form of Eq. 39. $\qquad\square$

## E. Proof of the Quantum Universal Approximation Theorem for Operator

We will often use the following notations.

$K$: some compact set in a Banach space.

$C(K)$: Banach space of all continuous functions defined on $K$, with norm $\|f\|_{C(K)} = \max_{x \in K} |f(\mathbf{x})|$.

$C_p[-1,1]^n$: All 2-periodic functions with period 2 with respect to every variable $x_i, i = 1, \cdots, n$.

**Lemma E.1.** *Suppose that $K$ is a compact set in $\mathbb{R}^n$, $f \in C(K)$, then there is a continuous function $E(f) \in C(\mathbb{R}^n)$, such that (1) $f(\mathbf{x}) = E(f)(\mathbf{x})$ for all $x \in K$; (2) $\sup_{x \in \mathbb{R}^n} |E(f)(\mathbf{x})| \leq \sup_{x \in K} |f(\mathbf{x})|$; (3) there is a constant $c$ such that*

$$\sup_{|\mathbf{x}'-\mathbf{x}''|<\delta} |E(f)(\mathbf{x}') - E(f)(\mathbf{x}'')| \leq c \sup_{|\mathbf{x}'-\mathbf{x}''|<\delta, \mathbf{x}',\mathbf{x}''\in K} |f(\mathbf{x}') - f(\mathbf{x}'')| \tag{45}$$

*Proof.* The proof of Lemma E.1 can be found in Stein (p.175). $\qquad\square$

**Lemma E.2.** *$V$ is a compact set in $C(K)$ if and only if*

1. *$V$ is a closed set in $C(K)$,*

2. *there is a constant $M$ such that $\|f(\mathbf{x})\|_{C(K)} \leq M$ for all $f \in V$,*

3. *$V$ is equicontinuous, i.e., for any $\epsilon > 0$, there exists $\delta > 0$ such that $|f(\mathbf{x}') - f(\mathbf{x}'')| < \epsilon$ for all $f \in V$, provided that $\mathbf{x}', \mathbf{x}'' \in K$ and $\|\mathbf{x}' - \mathbf{x}''\|_K < \delta$.*

*Proof.* The proof of Lemma E.2 can be found in Dieudonné (2011) (p.142). $\qquad\square$

**Lemma E.3.** *Suppose that $K$ is a compact set in $\mathbb{I}^n = [0,1]^n$, $V$ is a compact set in $C(K)$, then $V$ can be extended to a compact set in $C_p[-1,1]^n$.*

*Proof.* By Lemmas E.1 and E.2, $V$ can be extended to be a compact set $V_1$ in $C[0,1]^n$. Now, for every $f \in V_1$, define an even extension of $f$ as follows:

$$f^*(x_1, \cdots, x_k, \cdots, x_n) = f(x_1, \cdots, -x_k, \cdots, x_n) \tag{46}$$

Then $U \in \{f^* : f \in V_1\}$ is the required compact set in $C_p[-1,1]^n$. □

**Lemma E.4.** *Suppose that $U$ is a compact set in $C_p[-1,1]$,*

$$B_R(f; \mathbf{x}) = \sum_{|\mathbf{m}| \leq R} (1 - \frac{|\mathbf{m}|^2}{R^2})^\alpha c_{\mathbf{m}}(f) e^{i\pi\mathbf{m}\cdot\mathbf{x}} \tag{47}$$

*is the Bochner-Riesz means of Fourier series of $f$, where $\mathbf{m} = [m_1, \cdots, m_n]$, $|\mathbf{m}|^2 = \sum_{i=1}^n |m_i|^2$, $c_{\mathbf{m}}(f)$ are Fourier coefficients of $f$, then for any $\epsilon > 0$, there is $R > 0$ such that*

$$|B_R(f; \mathbf{x}) - f(\mathbf{x})| \leq \epsilon \tag{48}$$

*Proof.* The proof of Lemma E.4 can be found in Stein & Weiss (1971). □

**Lemma E.5.** *Suppose that $|\psi(f)\rangle$ is a continuous $m$-dim state function defined on the Banach space $C(K)$, $|\phi(\mathbf{x})\rangle$ is a continuous $m$-dim state function defined on the compact set $K$, there exist a real constant $c$, $(m+2)$-dim normalized state functions $|\Psi(f)\rangle$ and $|\Phi(\mathbf{x})\rangle$, such that*

$$\langle\psi(f)|\phi(\mathbf{x})\rangle = c\langle\Psi(f)|\Phi(\mathbf{x})\rangle \tag{49}$$

*Proof.* Without loss of generality, we can write $|\psi(f)\rangle$ and $|\phi(\mathbf{x})\rangle$ as

$$|\psi(f)\rangle = \sum_{i=1}^m g_i(f)|i\rangle, \quad |\phi(\mathbf{x})\rangle = \sum_{i=1}^m h_i(\mathbf{x})|i\rangle. \tag{50}$$

Since their amplitudes are both bounded, we define the 1-norm of the state function $\||\psi(f)\rangle\| = \max \|g_i(f)\|_1$. Because the case of $\||\psi(f)\rangle\| = 0$ or $\||\phi(\mathbf{x})\rangle\| = 0$ is trival, we can write $|\Psi(f)\rangle$ and $|\Phi(\mathbf{x})\rangle$ as:

$$
\begin{aligned}
|\Psi(f)\rangle &= \frac{1}{\||\psi(f)\rangle\|}(\sum_{i=1}^m g_i(f)|i\rangle + g_{m+1}(f)|m+1\rangle) \\
|\Phi(\mathbf{x})\rangle &= \frac{1}{\||\phi(\mathbf{x})\rangle\|}(\sum_{i=1}^m h_i(\mathbf{x})|i\rangle + h_{m+2}(\mathbf{x})|m+2\rangle)
\end{aligned} \tag{51}
$$

Since the norms of the first $m$ dimensions are less than or equal to 1, there exist $f_{i+1}$ and $g_{i+2}$ to normalize $|\Psi(f)\rangle$ and $|\Phi(\mathbf{x})\rangle$. Thus exist a real constant $c$, such that

$$\langle\psi(f)|\phi(\mathbf{x})\rangle = c\langle\Psi(f)|\Phi(\mathbf{x})\rangle \tag{52}$$

□

**Theorem E.6.** *Suppose that $K$ is a compact set in $\mathbb{R}^n$, and $U$ is a compact set in $C(K)$, then for any $\epsilon > 0$, there exist a positive integer $N$, a real constant $c$, a $N$-dim state function $|t\rangle$, which is independent of $f \in C(K)$, and a $N$-dim state functional $|b\rangle$ depending on $f$, such that*

$$|f(\mathbf{x}) - c\langle b(f)|t(\mathbf{x})\rangle| \leq \epsilon \tag{53}$$

*holds for all $x \in K$ and $f \in U$.*

*Proof.* Without loss of generality, we assume that $K \subseteq [0,1]^n$. By Lemma E.3, we can assume that $K = [-1,1]^n$ and $U \subseteq [-1,1]^n$. By Lemma E.4, for any $\epsilon > 0$, there exists $R > 0$, such that for any $\mathbf{x} = [x_1, \cdots, x_n] \in [-1,1]^n$ and $f \in U$, there holds:

$$| \sum_{|\mathbf{m}| \leq R} (1 - \frac{|\mathbf{m}|^2}{R^2})^\alpha c_{m_1 \cdots m_n}(f^*) \exp(i\pi(m_1 x_1 + \cdots + m_n x_n)) - f^*(x_1, \cdots, x_n)| < \epsilon \tag{54}$$

By the definition of the Fourier coefficients and the evenness of $f(\mathbf{x})$, we can rewrite it as:

$$| \sum_{|\mathbf{m}| \leq R} d_{m_1 \cdots m_n}(f^*) \cos(\pi(m_1 x_1 + \cdots + m_n x_n)) - f^*(x_1, \cdots, x_n)) - f^*(x_1, \cdots, x_n)| < \epsilon \tag{55}$$

where $d_{m_1 \cdots m_n}$ are real numbers. It is obvious that for every $\mathbf{x} \in [-1,1]^n$, the first item is the inner product of two states:

$$\sum_{|\mathbf{m}| \leq R} d_{m_1 \cdots m_n}(f^*)|m\rangle$$
$$\sum_{|\mathbf{m}| \leq R} \cos(\pi(m_1 x_1 + \cdots + m_n x_n))|m\rangle \tag{56}$$

By Lemma E.5, exist a positive intege $N$, a real constant $c$, a $N$-dim state functional $|b(f^*)\rangle$ and a $N$-dim state function $|t(\mathbf{x})\rangle$, such that

$$|f^*(\mathbf{x}) - c\langle b(f^*)|t(\mathbf{x})\rangle| < \epsilon \tag{57}$$

is true for all $x \in [-1,1]^n$ and $f^* \in V$. Thus

$$|f(\mathbf{x}) - c\langle b(f)|t(\mathbf{x})\rangle| < \epsilon \tag{58}$$

is true for all $x \in [0,1]^n$ and $f \in U$. $\qquad\square$

**Lemma E.7.** *$\{\eta_i\}_{i=1}^\infty$ is a sequence such that $\eta_1 > \eta_2 > \cdots \eta_n \to 0$ and $\{n(\eta_i)\}_{i=1}^\infty$ is a sequence of positive integers such that $n(\eta_1) < n(\eta_2) < \cdots < n(\eta_k) \to \infty$, such that the first $n(\eta_k)$ elements $N(\eta_k) = \{x_1, \cdots, x_{n(\eta_k)}\}$ is an $\eta_k$-net in K. We define functions*

$$T^*_{\eta_k, j}(\mathbf{x}) = \begin{cases} 1 - \dfrac{\|\mathbf{x} - \mathbf{x}_j\|_k}{\eta_k} & if \|\mathbf{x} - \mathbf{x}_j\|_X \leq \eta_k \\ 0 & otherwise \end{cases} \tag{59}$$

*and*

$$T_{\eta_k, j}(\mathbf{x}) = \frac{T^*_{\eta_k, j}(\mathbf{x})}{\sum_{j=1}^{n(\eta_k)} T^*_{\eta_k, j}(\mathbf{x})} \tag{60}$$

*for $j = 1, \cdots, n(\eta_k)$. For each $u \in V$, we define a function*

$$u_{\eta_k}(\mathbf{x}) = \sum_{j=1}^{n(\eta_k)} u(x_j) T_{\eta_k, j}(\mathbf{x}) \tag{61}$$

*and sets $V_{\eta_k} = \{u_{\eta_k} : u \in V\}$ and $V^* = V \bigcup (\bigcup_{k=1}^\infty V_{\eta_k})$. We then have the following results:*

*1) For each fixed $k$, $V_{\eta_k}$ is a compact set in a subspace of dimension $n(\eta_k) \in C(K)$.*

*2) For every $u \in V$, there holds*

$$\|u - u_{\eta_k}\|_{C(K)} \leq \delta_k \tag{62}$$

*3) $V^*$ is a compact set in $C(K)$.*

*Proof.* The proof of Lemma E.7 can be found in Chen & Chen (1995). $\qquad\square$

**Theorem E.8.** *Suppose that $X$ is a Banach Space, $K \subseteq X$ is a compact set, $V$ is a compact set in $C(K)$, $f$ is a continuous functional defined on $V$, then for any $\epsilon > 0$, there exist a positive integer $N$, a real constant $c$, a $N$-dim state function $|t\rangle$, which is independent of $f \in C(K)$, and a state functional $|b\rangle$ depending on $f$, such that*

$$|f(u) - c\langle b(f)|t(\mathbf{x})\rangle| \leq \epsilon \tag{63}$$

*holds for all $u \in V$.*

*Proof.* By Tietze's Extension Theorem, we can define a continuous functional on $V^*$ such that

$$f^*(u) = f(u) \quad \text{if } u \in V \tag{64}$$

Because $f^*$ is a continuous functional defined on the compact set $V^*$, therefore for any $\epsilon > 0$, we can find a $\delta > 0$ such that $|f^*(u) - f^*(v)| < \epsilon/2$ provided that $u, v \in V^*$ and $\|u - v\|_{C(K)} < \delta$. Let $k$ be fixed such that $\delta_k < \delta$, then by proposition 2. of Lemma E.7 for every $u \in V$,

$$\|u - u_{\eta_k}\|_X < \delta_k \tag{65}$$

which implies

$$|f^*(u) - f^*(u_{\eta_k})| < \epsilon/2 \tag{66}$$

for all $u \in V$. By proposition 1. of Lemma E.7, we see that $f^*(u_{\eta_k})$ is a continuous functional defined on the compact set $V_{\eta_k}$ in $\mathbb{R}^{n(\eta_k)}$. By Theorem E.6, we can find $N, c, |b\rangle, |t\rangle$, such that

$$|f^*(u_{\eta_k}) - c\langle b(f^*)|t(\mathbf{x})\rangle| < \epsilon/2 \tag{67}$$

We conclude that

$$|f(u) - c\langle b(f)|t(\mathbf{x})\rangle| < \epsilon \tag{68}$$

Thus, Theorem E.8 is proved. $\qquad\square$

**Theorem E.9.** *Suppose that $X$ is a Banach Space, $K_1 \subseteq X$, $K_2 \subseteq \mathbb{R}^n$ are two compact sets in $X$ and $\mathbb{R}^n$ respectively, $V$ is a compact set in $C(K_1)$, $G$ is a nonlinear continuous operator, which maps $V$ into $C(K_2)$, then for any $\epsilon > 0$, there exist a positive integer $N$, a real constant $c$, a $N$-dim state function $|t\rangle$ and a $N$-dim state functional $|b\rangle$, such that*

$$|G(u)(\mathbf{y}) - c\langle b(u)|t(\mathbf{y})\rangle| < \epsilon \tag{69}$$

*Proof.* From the assumption that $G$ is a continuous operator which maps a compact set $V$ in $C(K_1)$ into $C(K_2)$, it is straightforward to prove that the range $G(V) = \{G(u) : u \in V\}$ is also a compact set in $C(K_2)$. By Theorem E.6, for any $\epsilon > 0$, there are a positive integer $N$, a real constant c, a $N$-dim state function $|t\rangle$ and a $N$-dim state functional $|b\rangle$, such that

$$|G(u)(\mathbf{y}) - c\langle b(G(u))|t(\mathbf{y})\rangle| < \epsilon \tag{70}$$

holds for all $y \in K_2$ and $u \in V$.

Since $G$ is a continuous operator, combining with the last proposition of Theorem E.6, we conclude that for each $k = 1, \cdots, N$, $|b(G(u))\rangle$ is a continuous state functional defined on $V$.

$$|G(u)(\mathbf{y}) - c\langle b(u)|t(\mathbf{y})\rangle| < \epsilon \tag{71}$$

$\qquad\square$

# F. Extensive benchmarking

*Table 6.* Test error comparison of TF-QuanONet with different qubits and params. #, where all Trunk Depth is set to 10 and Ansatz Depth is set to 2. Under the condition of different qubit numbers, the performance of TF-QuanONet generally increases with the increase of Branch Depth. Interestingly, our method performs better at low qubits, which may be because the deeper Branch Net improves the expressivity in the frequency domain.

| Qubits # | Branch Depth | MSE | Param. # | Epochs # |
|---|---|---|---|---|
| | 50 | 6.5e-5 | 960 | 1000 |
| 2 | 100 | 5.8e-5 | 1760 | 1000 |
| | 150 | 4.9e-5 | 2560 | 1000 |
| | 25 | 4.4e-5 | 1120 | 1000 |
| 4 | 50 | 3.2e-5 | 1920 | 1000 |
| | 75 | 3.7e-5 | 2720 | 1000 |
| | 20 | 2.2e-4 | 1200 | 1000 |
| 5 | 40 | 1.7e-4 | 2000 | 1000 |
| | 60 | 1.3e-4 | 2800 | 1000 |
| | 10 | 3.15e-4 | 1600 | 1000 |
| 10 | 20 | 1.45e-4 | 2400 | 1000 |
| | 30 | 1.05e-4 | 3200 | 1000 |

*Table 7.* Execution time of TF-QuanONet on a quantum simulator and DeepONet with 10000 training instances based on different frameworks. The forword time of TF-QuanONet is about 10-100 times that of DeepONet at the same scale

| Framework | TF-QuanONet | | DeepONet | |
|---|---|---|---|---|
| | Training | Inference | Training | Inference |
| DeepXde | 82.1543 | 0.000059858s | 0.8119s | 0.000000414s |
| Torch | 168.0444s | 0.000082172s | 0.9917s | 0.000000411s |
| MindSpore | 10.4797s | 0.000008763s | 1.5054s | 0.000000467s |

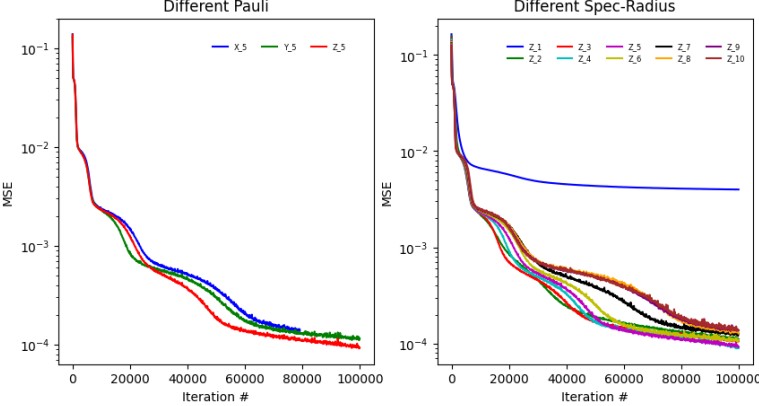

*Figure 12.* Error curve of TF-QuanONet with different Hamiltonian for anti-derivative operator, Z-7 means $H = \frac{7}{5} \sum_{i=1}^{5} \sigma_z^i$ with spectral radius equals 7. Further increasing the spectral radius has little impact on the results, but too small spectral radius will limit the range of the solution function and thus affect the prediction accuracy.

