# OpenReview forum: "QuanONet: Quantum Neural Operator with Application to Differential Equation"
_ICML.cc/2025/Conference — ICML 2025 poster_

### Official Review · Reviewer_xycX · 2025-03-08

**Overall Recommendation:** 3

**Summary:**

This paper proposes QuanONet, A quantum analogy of neural operators, which can be executed in quantum computers. The paper extends the classical universal approximation theorem. The proposed architecture QunONet retains powerful generalization of classical neural operators. The work also highlights a version called TF-QuanONet based on the trainable-frequency method, reducing the requirement of deep quntum circut repetition.

## update after rebuttal

After carefully reading the rebuttal and other reviews, I decided to keep my original score. The authors are encouraged to incorporate suggested updates in the revision.

**Claims And Evidence:**

The main claims presented in this paper are versatility and scalability incorporating advancements of DeepONet, such as Branch Net and Trunk Net.  With unique architectural properties for QNNs such as hardware efficient encoder, entangle, and ansatz layers the proposed architecture was able to outperform other quantum methods. Compared to other QNNs, such as QFNO and Quantum DeepONet, the authors argue that QuanONet truly integrates the spirit of QNNs into the core architecture applicable to quantum computing machines. It reads that this property enables the authors to extend QNN-related approximation theories (Theorems 2.1 & 2.2) to Quantum State Functions (Theorem 3.1). Lastly, this facilitates of TF-QuanONet, which overcomes the difficulty of setting coefficients and improve the robustness.

**Essential References Not Discussed:**

Essential references are well-discussed in this paper.

**Experimental Designs Or Analyses:**

Experiments are based on four types of ODE and PDE problems. Based on the results QuanONet shows best results depending on the hyperparameter $\lambda$. Since I am not expert in this field, experiment results for QuanONet seems to be quite good but I am not fully confident on this initial assessment. Also, I have skepticism for TF-QuanONet since they are strictly worse than QuanONet. Therefore, performance of QuanOnet might be mainly due to extensive architecture and hyperparameter searches by the authors before the actual experiments.

**Methods And Evaluation Criteria:**

Benchmark data such as antiderivative operators and diffusion-reaction systems are synthetically generated, and can be replicated by the main paper and supplementary materials.

**Other Comments Or Suggestions:**

The formatting style of Figures 4 and 5 is not good; I suggest to increase the font size of numbers and labels. Regarding the conclusion "How to compare QuanONet and classical neural operators is a controversial issue.", I do not agree with this opinion. As the QNNs are capable of modeling various ODEs and PDEs, the authors are encouraged to point out actual performance comparisons and discuss the particular reason QNNs are not as efficient as classical neural operators.

**Other Strengths And Weaknesses:**

In my reading, the explanation of TF-QuanONet is not sufficient for me to understand. Furthermore---if I understand the results correctly---TF-QuanONet strictly performs worse than QuanONet("error evolution of TF-QuanONet is poor," and "shows obvious overfitting phenomenon.") Even tough, TF-QuanONet reduces the burden of setting coefficients, I think TF-QuanOnet does not have noticeable benefits. This raises the question of presenting claims TF-QuanOnet as in the introduction section.

**Questions For Authors:**

* What is the computational complexity (perhaps execution time) of QuanONet, TF-QuanONet and other baselines?

**Relation To Broader Scientific Literature:**

This work could bridge to gab between machine learning and natural science, especially particle physics. There has been multiple ways of simulating quantum systems, and QuanONet might be beneficial for finding scientific breakthrough if this is combined with broader scientific literature.

**Theoretical Claims:**

The theoretical claim of the paper is the justification of the proof of Theorem 3.1 seems to rely on the universal Hamiltonian family. Theorem 3.2 deals with Quantum Universal Approximation. Judging by the brevity of Appendix E, I am not fully convinced whether these theorems show how the architectural advancements presented in this works is justified. Therefore, I would say some theoretical results are properly placed, but I think the implication of the theory does not fully cover every implementation details of QuanONet architectures.

---

> ### Author Rebuttal · Authors · 2025-04-01
>
> > Judging by the brevity of Appendix E, I am not fully convinced whether these theorems show how the architectural advancements presented in this works is justified.
>
> We apologize for the confusion caused by the over-brevity of the theory and add a detailed proof of a stronger version of quantum universal approximation theorem for operators with the form of $|f(x)-c\langle b(f)|t(x\rangle|\leq \epsilon$ and rigorously address continuity and compactness in Banach spaces. Due to space limitations in Rebuttal, we would like to ask if it is possible to provide the details of the proof in pictorial form?
>
> > The explanation of TF-QuanONet is not sufficient for me to understand. Furthermore---if I understand the results correctly---TF-QuanONet strictly performs worse than QuanONet("error evolution of TF-QuanONet is poor," and "shows obvious overfitting phenomenon.") Even tough, TF-QuanONet reduces the burden of setting coefficients, I think TF-QuanOnet does not have noticeable benefits. This raises the question of presenting claims TF-QuanOnet as in the introduction section.
>
>  In initial experiments, the 2k training instances was too small to generalize. We provide the results of five runs on a much larger dataset with batch size 100 (Training instances: ODE 10k, PDE 100k; Testing instances:  ODE 100k, PDE 1000k), aligning all methods for 100K iterations, as a more convincing setting.  The results can be seen in our first response to reviewer CFgz and fourth response to reviewer LZaG.  TF-QuanONet outperforms other quantum  methods on all problems and is superior to classical methods except for nonlinear operators. Its performance almost  does not depend on coefficient selection.
>
> > The formatting style of Figures 4 and 5 is not good; I suggest to increase the font size of numbers and labels.
>
>  Thanks for your suggestions, we have updated the figure in the new link.
>
> > What is the computational complexity (perhaps execution time) of QuanONet, TF-QuanONet and other baselines?
>
>   For QNNs, params-shift method is widely used for gradient calculation, that is, the gradient of the parameters is obtained by two measurements. Therefore, the training complexity of QNNs mainly depends on the number of parameters and the selection of observations. This is all aligned in our experiments. More discussion of the complexity of the measurements can also be found in our fourth response to reviewer Eswa.
> The comparison between classical and quantum methods is not easy, especially since current quantum computers require more shots to mitigate noise. We provide a comparison of training and inference time (averaged over 1e6 iterations on 10k training instances and 1e7 inferences) between quantum simulators (implementing TF-QuanONet with classical computation) and DeepONet under different frameworks as shown in [Tab. 5](https://anonymous.4open.science/r/ICML-2025-rebuttal-42ED/Table5.jpg).
>
>
>
> > Regarding the conclusion "How to compare QuanONet and classical neural operators is a controversial issue.", I do not agree with this opinion. As the QNNs are capable of modeling various ODEs and PDEs, the authors are encouraged to point out actual performance comparisons and discuss the particular reason QNNs are not as efficient as classical neural operators.
>
>  Thanks for your suggestion. We further add new comparison between QNN and classic operators, and give more discussion based on the new results. Please also check our first response to reviewer CFgz and fourth response to reviewer  LZaG for specific new experiments and discussion which we will add to our revisio.
> For our sentence, we will remove it and give a more specific discussion based on our new results.

---

### Official Review · Reviewer_Eswa · 2025-03-10

**Overall Recommendation:** 3

**Summary:**

This paper proposes a new model namely QuanONet (and TF-QuanONet) which is the first model purely based on quantum circuit. The paper further generalizes the approximation theorem of DeepONet to quantum setting. It has experiments on ODEs and PDEs and it shows advantages of QuanONet compared to previous quantum neural networks.

**Claims And Evidence:**

I think the quantum neural operator needs more motivation -- Are quantum methods fundamentally faster for operator learning problems? For example, for which type of PDEs do we expect quantum methods to be faster?

Meanwhile, the authors mentioned noisy intermediatescale quantum (NISQ) and fault-tolerance in the introduction. It seems not supported in the experiments.

**Essential References Not Discussed:**

N/A

**Experimental Designs Or Analyses:**

N/A

**Methods And Evaluation Criteria:**

The paper contains experiments on operator learning consisting of antiderivative operator, simple ODE, and 1-dimensional PDE. These problems can be perfectly solved using classical methods such as numerical solver and classical ML models. It would be more interesting to discuss which problem quantum method can potentially show advantage, and add experiments on these.

Besides, while previous works such as Quantum DeepONet and Quantum FNO have classical components, it is still very interesting to add these baselines to see the gap of pure quantum and hybrid methods.

**Other Comments Or Suggestions:**

Comments:
- It would be helpful to add Quantum DeepONet and Quantum FNO as baseline models.
- Figure 4 does not provide much information. A table should be sufficient.

**Other Strengths And Weaknesses:**

While the motivation can be improved, the paper show potential to design neural operator using pure quantum circuits.

**Questions For Authors:**

- What PDEs does quantum methods have an advantage? Maybe high-dimension Schrodinger equation?
- How many Qbit we would need to run this model?
- if somehow we get exponential speedups using quantum method, how do we extract the solution? Is the observation costly?

**Relation To Broader Scientific Literature:**

N/A

**Theoretical Claims:**

The paper contains an approximation theorem for quantum neural operator. The theorem looks reasonable to me. Whilemean, the proof seems a translation of linear algebra into quantum setting.

---

> ### Author Rebuttal · Authors · 2025-04-01
>
> > How many Qbits we would need to run this model?
>
> Based on the experience of DeepONet, operator problems can be solved well by taking vector dimensions as 10-1000, so only less than 10 qubits are needed to build QuanONet. More discussion is visible in our first response to reviewer LZaG. Notably, our method performs well even with only two qubits (MSE<1e-4).
>
> > What PDEs does quantum methods have an advantage? Maybe high-dimension Schrodinger equation?
>
> Quantum methods exhibit particular advantages for operator problems at low frequencies, as the spectral lines can be effectively captured through finite-depth circuit implementations (see our detailed analysis in the first and seventh responses to reviewer CFgz).
> For higher-dimensional PDE, since the operator approximation theorem requires input function discretization, they all face fundamental limitations of exponential input dimension.
>
> > It would be helpful to add Quantum DeepONet and Quantum FNO as baseline models.
>
>
> Since Quan-DeepONet only utilizes quantum circuits to speed up the linear layer of DeepONet inference, the same classical network setup is used for the nonlinear layer as well as the training part, so it's unable to achieve better results than DeepONet (or even worse) [Xiao P, et al. Quantum DeepONet: Neural operators accelerated by quantum computing](https://arxiv.org/pdf/2409.15683), so we provide results for DeepONet as an alternative.
>
> In the existing research on Quan-FNO, no advantage over FNO on operator problems is observed, and more than 100 qubits are needed, which exceeds the limitations of all Quantum simulators and quantum real machines, as shown in [Tab. 3](https://anonymous.4open.science/r/ICML-2025-rebuttal-42ED/Table3.jpg). (Quan-)FNO is only applicable to aligned data (i.e., sampling all sensors for each initial function) while QuanONet and DeepONet do not have such restrictions.
>
> We provide the results of FNO with 100 initial functions as training instances, as shown in [Tab. 4](https://anonymous.4open.science/r/ICML-2025-rebuttal-42ED/Table4.jpg)
>
>
>
> > Meanwhile, the authors mentioned noisy intermediatescale quantum (NISQ) and fault-tolerance in the introduction. It seems not supported in the experiments.
>
> The hardware noise characteristics in the NISQ era are primarily influenced by qubit count, gate cost and circuit depth. We conducted extensive benchmarking across various qubit count and layer depths, with complete experimental results and analysis presented in the first response to reviewer LZaG2. Remarkably, TF-QuanONet demonstrates exceptional precision (MSE <1e-4) even with 2 qubits. Moreover, QuanONet's hardware-friendly circuit design philosophy and ultra-low qubit requirements enable full exploitation of modern quantum compilation optimizations, which result in a significant reduction in circuit depths.
>
> The [Fig. 5](https://anonymous.4open.science/r/ICML-2025-rebuttal-42ED/Fig5.jpg) further supports the performance of our method on real-devices. We trained a 2-qubits TF-QuanONet for antiderivative tasks and tested it on IBM brisbane Q57/Q58 (T1/T2 = 252/178 μs, ECR error = 7.9e-3, readout error = 1.51e-2). By leveraging Qiskit's compilation optimization techniques, the circuit depth was reduced to merely 20 layers. Using y=x with 100 points in [0,1] and standard noise mitigation (gate twirling, ZNE, TREX), we achieved MSE = 1.57e-3 (vs. 1.5e-5 simulation). The gap is mainly attributed to non-ideal gate operations and residual decoherence effects.
>
> QuanONet shows the high cost-efficiency in utilizing the limited and currently available qubit resource. This feature is especially welcomed along the FTQC trend as recently discussed in the Nature paper: [R. Acharya et al., Quantum error correction below the surface code threshold](https://doi.org/10.1038/s41586-024-08449-y). Unlike the other QML areas like vision, which still requires large number of qubits for real-world data, We believe our technique is more promising for showing the value of QML.
>
> > if somehow we get exponential speedups using quantum method, how do we extract the solution? Is the observation costly?
>
> QuanONet extracts the solution by measuring the expectation value of a Hamiltonian composed of commuting single-qubit Z-terms and scaling the result.
>
>  The observation cost primarily stems from three factors: the number of measurement groups, shots per group, and error mitigation overhead.
>
> 1.Due to the Hamiltonian’s commuting structure, we can measure them simultaneously in a single group, avoiding the grouping overhead required for non-commuting observables.
>
> 2.The shots required to achieve precision $\epsilon$ scale as $O(n _{qubits}/\epsilon^2)$, where the numerator arises from the independent variances of the $Z$-terms, independent of problem dimension.
>
> 3.Experimental results on IBM brisbane demonstrate that $10^4$ shots suffice to achieve < 4e-2 absolute error, validating the practicality of our approach on NISQ hardware.

---

> > ### Comment · Reviewer_Eswa · 2025-04-02
> >
> > Thanks the authors for the response. It is impressive that the author can train a 2-qubits TF-QuanONet for antiderivative tasks and tested it on IBM. Well it is still very primitive, but it shows the potential. I will raise my score to 3.

---

> > > ### Author Response · Authors · 2025-04-02
> > >
> > > We are sincerely grateful for your careful reading and thoughtful comments, which have been invaluable in enhancing the clarity and rigor of our work. Thank you again for your time and effort in reviewing our paper.
> > >
> > > Best regards

---

### Official Review · Reviewer_LZaG · 2025-03-14

**Overall Recommendation:** 4

**Summary:**

The paper introduces QuanONet, a quantum neural operator framework designed to solve differential equations using pure quantum circuits. The authors extend classical universal approximation theorems to quantum settings, proving that quantum neural networks (QNNs) can approximate operators for differential equations. They propose two architectures: QuanONet, a hardware-efficient quantum neural operator, and TF-QuanONet, which incorporates trainable frequencies to improve robustness. Experiments on antiderivative operators, homogeneous/nonlinear ODEs, and a diffusion-reaction PDE demonstrate that QuanONet outperforms existing quantum methods and competes with classical baselines like DeepONet in certain cases. The theoretical foundation, empirical results, and focus on NISQ-compatibility position QuanONet as a novel contribution to quantum machine learning for operator learning.

**Claims And Evidence:**

- **Claim:** QuanONet is the first pure quantum neural operator.
  **Evidence:** The architecture uses only quantum circuits without classical components, differentiating it from hybrid approaches like QFNO. This is supported by the circuit design in Fig. 2.
- **Claim:** TF-QuanONet improves robustness by dynamically adjusting frequency spectra.
  **Evidence:** Table 1 and Fig. 4 show TF-QuanONet performs consistently across coefficient settings, unlike QuanONet.
- **Claim:** QuanONet outperforms classical methods like FNN and DeepONet.
  **Evidence:** Results in Table 1 show lower errors for some tasks, but comparisons are limited by differing parameter scales and training iterations (e.g., 100 vs. 10,000 iterations for PDEs). This claim requires further validation under matched computational budgets.

**Essential References Not Discussed:**

None

**Experimental Designs Or Analyses:**

- **Training Data Size:** Only 2,000 samples for ODEs may be insufficient for robust generalization.
- **Coefficient Sensitivity:** QuanONet’s performance heavily depends on $\(\lambda\)$ (Fig. 4), raising concerns about practicality.
- **PDE Experiment:** The 100-iteration limit for QuanONet vs. 10,000 for DeepONet undermines claims of efficiency. A fair comparison requires equal computational effort.

**Methods And Evaluation Criteria:**

- **Strengths:** The use of Gaussian random fields for data generation aligns with prior work (Lu et al., 2021). The focus on NISQ-compatible hardware-efficient ansatzes is pragmatic.
- **Weaknesses:**
  - Parameter counts are matched across methods, but quantum vs. classical architectures are fundamentally different, making direct comparisons less meaningful.
  - The choice of 5 qubits and fixed Hamiltonian $\(H = \sum \sigma_z\)$ may limit exploration of quantum advantages.
  - Training iterations for PDEs (100 for QuanONet vs. 10,000 for DeepONet) skew performance comparisons.

**Other Comments Or Suggestions:**

None

**Other Strengths And Weaknesses:**

None

**Questions For Authors:**

1. **Theoretical Proofs:** Could you provide a more detailed construction of $\(W\)$ and $\(|\Gamma\rangle\)$ in Theorem 3.1 to clarify how arbitrary states are approximated?
   *Impact:* A clearer proof would strengthen theoretical claims.
2. **Training Iterations:** Why are training iterations for PDEs vastly different between QuanONet (100) and DeepONet (10,000)? Would QuanONet maintain its advantage with equal iterations?
   *Impact:* If QuanONet’s performance degrades with more iterations, claims about efficiency would weaken.
3. **Hardware Constraints:** How does QuanONet address limited qubit connectivity or noise in real NISQ devices?
   *Impact:* Practical applicability hinges on robustness to hardware limitations.

**Relation To Broader Scientific Literature:**

The paper situates QuanONet within quantum methods for differential equations (e.g., QFNO, quantum DeepONet) and classical neural operators (DeepONet). However, it does not engage with recent hybrid quantum-classical operator approaches (e.g., Smith et al., 2023) that combine classical neural networks with QNNs, which could provide valuable context.

**Theoretical Claims:**

- **Theorem 3.1 (Quantum Universal Approximation for State Functions):** The proof in Appendix D relies on Fourier series approximations, building on Schuld et al. (2021). While plausible, the proof lacks detailed steps for critical transitions (e.g., constructing $\(W\)$ and \$(|\Gamma\rangle\))$.
- **Theorem 3.2 (Quantum Universal Approximation for Operators):** The proof in Appendix E maps classical DeepONet structures to quantum states but does not rigorously address continuity or compactness in Banach spaces. A more formal treatment is needed.

---

> ### Author Rebuttal · Authors · 2025-04-01
>
> > The choice of 5 qubits and fixed Hamiltonian may limit exploration of quantum advantages.
>
> We choose the number of qubits to be 5 mainly based on the experience of DeepONet that operator problems can be solved well by taking vector dimensions as 10-1000, so the quantum state dimension $2^5=32$ can balance accuracy and efficiency. This is also an acceptable number of bits for existing quantum devices.
> The choice of the Hamiltonian does make sense. In our attempts, further increasing the spectral radius has little impact on the results, but too small spectral radius will limit the range of the solution function and thus affect the prediction accuracy.
>
>
>
> Experimental results of TF-QuanONet across varying qubit counts and branch depths for the antiderivative operator (note: not all models reached full convergence due to time constraints) as shown in [Tab. 2](https://anonymous.4open.science/r/ICML-2025-rebuttal-42ED/Table2.jpg) and [Fig. 3](https://anonymous.4open.science/r/ICML-2025-rebuttal-42ED/Fig3.jpg)
>
>
> >  Quantum vs. classical architectures are fundamentally diferent, making direct comparisons less meaningful.
>
>  The training efficiency of QNN is indeed currently not comparable to NN, so we focus on highlighting the advantages of QuanONet over other QNN frameworks. Although QuanONet experimentally performs better than classical methods when aligning parameters and number of iterations in most cases, it is currently still a common bottleneck to the community beyond our paper, for maintaining the performance on real quantum device due to the limited quantum hardware and noise. We will clarify this point in our revision.
>
>
> > Could you provide a more detailed construction of $W$ and $|\Gamma\rangle$ in Theorem 3.1 to clarify how arbitrary states are approximated?
>
>  We apologize for the confusion caused by the expression brevity to the theory. $W$ is realized by a quantum repeated parameter layer that the whole Hilbert space is completely controllable, that is, the generators of the used parameter quantum gates $g=\set{H_p}_{p=1}^P$ can span a dynamic Lie algebra(DLA) $\mathfrak g$ of dimension $4^n$ ($n$ is the number of qubits, the corresponding algebra is $SU(2^n)$).
>
> Theorem [Controllability of molecular systems. Physical Review A, 51(2):960, 1995].
> 1. All coherent superpositions of states can be achieved if $S$ equals $U(N)$. This is equivalent to requiring that $\mathfrak g$ be the Lie algebra of all $N\times N$ skew-Hermitian matrices, which in turn is equivalent to requiring that the dimension of $\mathfrak g$ as a vector space over the real numbers is precisely $N^2$. The latter two of the above equivalent conditions are also necessary for controllability.
> 2. All probability amplitudes can be achieved if $S$ is compact and contains $SU(N)$ which is equivalent to demanding that $\mathfrak g$ is the Lie algebra of all $N\times N$ skew-Hermitian matrices. In particular, if all probability amplitudes can be achieved then one can obtain all coherent superpositions of states.
>
> Based on the above theorem, $W$ satisfying this property can approximate any $n$ bit quantum operator with ideal depth and  $|\Gamma\rangle=W|0\rangle$ which is an initial state acted by $W$ can approximate any $n$ bit quantum state. We adopt HEA as the structure of Branch and Trunk layers with {Rz, Ry, CNOT} that act on any qubit (or near-connected qubits) to span $SU(2^n)$. But purely one type of Pauli rotation gate or RBS gate does not conform (generators of RBS gates only span a subalgebra of $SO(2^n)$).
>
> The detailed discussion of Theorem 3.2 can be found in the first reply torReviewer xycX.
>
> > Why are training iterations for PDEs vastly diferent between QuanONet (100) and DeepONet (10,000)? Would QuanONet maintain its advantage with equal iterations?
>
> Our added experiments in the first of our response to reviewer CFgz, provide the results of five runs on a much larger dataset, aligning all methods for 100K iterations, as a more convincing setting.  Our approach achieves robust generalization.
> Taking the antiderivative operator as an example, the loss curves of all methods are as [Fig. 4](https://anonymous.4open.science/r/ICML-2025-rebuttal-42ED/Fig4.jpg). Here the (TF-)QuanONet's performance improves as the number of iterations increases more intuitively.
>
> >  How does QuanONet address limited qubit connectivity or noise in real NISQ devices?
>
> The two-bit quantum gates in QuanONet are all nearest neighbor connected. Due to its hardware efficient design, it can benefit from a very small number of high-quality qubits, which fits well with the current trend from NISQ to FTQC era. We provide complementary experiments on IBM brisbane to further precisely this point, and detailed results can be found in the fourth reply to reviewer Eswa.

---

### Official Review · Reviewer_CFgz · 2025-03-14

**Overall Recommendation:** 3

**Summary:**

The paper introduces QuanONet, a quantum neural network framework for learning nonlinear operators in differential equations. The primary contribution is extending classical universal approximation theorems for operators to quantum state versions. Two variants are proposed: a standard QuanONet with hardware-efficient pure quantum circuits, and TF-QuanONet featuring trainable frequency encoding to improve generalization and overcome coefficient selection challenges. Experimental evaluations demonstrate competitive performance compared to classical approaches and superior results relative to existing quantum methods on various differential equation problems.

**Claims And Evidence:**

The theoretical foundations presented in the paper are robust, with formal proofs (Theorems 3.1 and 3.2) providing sound theoretical grounding. Empirical results across several differential equation problems validate that QuanONet outperforms alternative quantum methods when properly configured, while TF-QuanONet demonstrates enhanced robustness across different initialization settings.

Claims regarding NISQ device applicability require additional substantiation, particularly concerning error propagation and mitigation in authentic quantum hardware environments. The comparison methodology with quantum approaches is methodologically sound, though comparisons with classical methods present limitations due to implementation differences.

**Essential References Not Discussed:**

The paper covers most relevant literature, although it could benefit from some more discussion of nnqs/vmc with symmetries, some recent pinn related works, and  barren plateau.

**Experimental Designs Or Analyses:**

The experimental design is generally appropriate but has some limitations. The experiments cover a range of differential equation problems with increasing complexity, and the authors test under different coefficient settings to demonstrate the robustness of their TF-QuanONet approach. The parameter counts are controlled across models for fair comparison, which strengthens the validity of the results.

However, the authors use simulated quantum environments rather than actual quantum hardware, which leaves questions about practical implementation challenges. The analysis of convergence issues with TF-QuanONet (mentioned on page 7) is somewhat superficial and would benefit from deeper investigation. The error evolution plots (Fig. 4) show interesting patterns that warrant more detailed analysis, especially the overfitting phenomenon mentioned for TF-QuanONet in Fig. 4(c). Furthermore, the performance gain over classical methods isn't consistently demonstrated across all cases, which weakens the overall impact.

**Methods And Evaluation Criteria:**

The experimental framework uses appropriate differential equation test cases with varying complexity levels.

However, the analysis primarily focuses on approximation error metrics without exploring training efficiency, hardware requirements, or scalability characteristics. While the selected test problems serve as effective benchmarks, they may not fully represent the complexities encountered in advanced scientific applications.

The convergence analysis of TF-QuanONet would benefit from expanded investigation, particularly regarding the overfitting phenomena observed in error evolution plots. The simulations are conducted in idealized quantum environments rather than actual quantum hardware, leaving implementation challenges unaddressed.

**Other Comments Or Suggestions:**

The paper would benefit from a more explicit discussion of limitations, particularly regarding current quantum hardware constraints. A clearer roadmap for how this work might lead to practical quantum advantage would strengthen the impact. More ablation studies on the effects of quantum circuit depth, Hamiltonian choice, and encoding strategies would provide valuable insights.

**Other Strengths And Weaknesses:**

The paper introduces a novel, theoretically-grounded approach to quantum neural operators, which represents a significant advance in the field. The trainable frequency technique is an innovative solution to the coefficient initialization problem, and the theoretical analysis connecting quantum circuits to Fourier series representations provides valuable insights. The pure quantum circuit design (without hybrid classical-quantum components) represents an advance over existing approaches.

Despite these strengths, several weaknesses should be addressed. The practical implementation on near-term quantum devices is not addressed sufficiently, leaving questions about the actual feasibility in NISQ-era hardware. The performance advantages over classical methods are inconsistent and depend heavily on hyperparameter settings. The paper lacks analysis of computational complexity or potential quantum advantages in terms of asymptotic scaling. The convergence challenges with TF-QuanONet mentioned briefly deserve more investigation. The D-R system case seems to be treated differently from the other examples with less detailed analysis.

**Questions For Authors:**

1. How would quantum hardware noise affect QuanONet performance, and what error mitigation strategies would be appropriate for practical implementation?
2. What mechanisms drive the overfitting phenomenon observed with TF-QuanONet, and what strategies might mitigate this behavior?
3. Beyond trainable frequencies, what alternative approaches might address the coefficient initialization challenge?
4. How does QuanONet relate to Neural Network Quantum States research, and could insights from that domain inform further development?
5. What specific properties of the diffusion-reaction system make it particularly amenable to quantum approaches?
6. Given the specific Hamiltonian requirements for different problems, how feasible would transfer learning or multi-task learning be with this architecture?

**Relation To Broader Scientific Literature:**

The paper properly positions itself at the intersection of quantum computing and neural operators for differential equations. The authors provide a comprehensive background on both quantum neural networks and neural operators, especially DeepONet. The comparison with related work is thorough, covering quantum algorithms for solving differential equations, universal approximation theorems for QNNs, and neural operator approaches. The distinction between existing hybrid quantum-classical approaches (like QFNO and quantum DeepONet) and their pure quantum approach is clearly articulated.

However, the paper could benefit from more discussion of how this work relates to broader quantum advantage questions, particularly given recent work on quantum neural network expressivity. Clearer explanations of how this approach compares to other quantum PDE solvers in terms of computational complexity, not just empirical performance, would strengthen the work.

**Theoretical Claims:**

The connection between theoretical principles and architectural implementation is clearly established, particularly in demonstrating how the frequency spectrum analysis informs the advantages of TF-QuanONet over the standard implementation. The theoretical formulation represents a significant strength of the paper. The extension from classical to quantum approximation theorems follows logical progression with well-structured proofs.

---

> ### Author Rebuttal · Authors · 2025-04-01
>
> > The performance gain over classical methods isn't consistently demonstrated across all cases, which weakens the overall impact.
>
> We add more experiments with larger scale with batch size 100, 100K max iterations and for five runs.
>
> 1) ODE: 10K train instances and 10K test instances.
> 2) PDE: 100K train instances and 100K test instances.
>
>  The results are given in [Tab. 1](https://anonymous.4open.science/r/ICML-2025-rebuttal-42ED/Table1.png). TF-QuanONet outperforms other quantum methods on all problems and is superior to classical methods except for nonlinear operators.
>
>
> >How would quantum hardware noise affect QuanONet performance, and what error mitigation strategies would be appropriate for practical implementation?
>
> Compared to other influential quantum algorithms, QuanONet enjoys the advantages of fewer qubits, lower demand for topological connectivity, and a simpler gate set that is easy to implement (e.g., avoiding complex multi-controlled gates). It can be further improved by noise mitigation (e.g. ZNE, MPS pretraining, and noise modeling), similar to [W. Sun, J. Xu, and C. Duan, Noise-Mitigated Variational Quantum Eigensolver with Pre-training and Zero-Noise Extrapolation](https://arxiv.org/abs/2501.01646).
>
> Besides, we implemented physical device testing on IBM brisbane using the y=x anti-derivative problem as benchmark with standard noise mitigation (gate twirling, ZNE, TREX). The hardware implementation achieved MSE = 1.57e-3 (vs simulation MSE = 1.5e-5), with residual error primarily attributable to non-ideal gate operations and residual decoherence effects (T1/T2 = 252/178 μs, ECR error = 7.9e-3, readout error = 1.51e-2). Extended details and nalysis are provided in our fourth response to reviewer Eswa.
>
>
> >  What mechanisms drive the overfitting phenomenon observed with TF-QuanONet, and what strategies might mitigate this behavior?
>
>  2k training instances in our initial submission is too small to fully reflect the frequency characteristics of the problems, thus TF-QuanONet mislearns the dominant frequency. We also find that DeepONet exhibits overfitting on these 2k training instances.
>  In added experiments, as the number of instances increases to 100K, the performance of TF-QuanONet is significantly improved and is much better than other quantum methods.
>
> > Beyond trainable frequencies, what alternative approaches might address the coefcient initialization challenge?
>
> We envision that the dominant frequency of the problem can be learned by a specialized small-scale TF-QNN. We do not care how the small-scale QNN performs, but rather provide a good coefficient setting strategy for the larger scale QuanONet through its trained coefficient distribution. For the time being, we leave it as future work due to the short rebuttal period.
>
>
> > How does QuanONet relate to Neural Network Quantum States research, and could insights from that domain inform further development?
>
>  It's an interesting association. QuanONet uses parameterized quantum state to construct Neural Operator (Quan4AI); NN Quantum state uses NN to learn quantum states (AI4Quan). They both embody the research promise of combining AI with scientific problems. We will add discussion in our final version.
>
> > What specific properties of the diffusion-reaction system make it particularly amenable to quantum approaches?
>
>  D-R system exhibits a lower frequency distribution compared to the other problems as shown in [Fig. 1](https://anonymous.4open.science/r/ICML-2025-rebuttal-42ED/Fig1.jpg).
>
> The frequency characteristics are closely related to the performance of QuanONet. For low frequency problems, QuanONet only needs a lower depth to achieve spectrum coverage. With this condition, it prioritises learning the dominant frequency of the problem (unlike NN learning from low frequencies) based on the frequency principle and hence more efficient. [Xu Y H, Zhang D B. Frequency principle for quantum machine learning via Fourier analysis.](https://arxiv.org/pdf/2409.06682)
>
> > Given the specific Hamiltonian requirements for diferent problems, how feasible would transfer learning or multi-task learning be with this architecture?
>
>  The choice of Hamiltonians are discussed detailly in the first response to reviewer LZaG.
>
> The input of neural operator can be extended to a tensor product of multiple functions (initial functions, driving terms, boundary conditions, etc.) to construct a multi-task learning method for differential equations. [Jin P, Meng S, Lu L. MIONet: Learning multiple-input operators via tensor product](https://arxiv.org/pdf/2202.06137) uses low rank approximation to avoid exponential dimension. For QuanONet, if a number of branch nets act independently on states $|b_i(u_i)\rangle$ and input functions $f_i$, the entire quantum system $|b_1(u_1)\rangle\otimes\cdots\otimes |b_n(u_n)\rangle$ is naturally in tensor product form, as shown in [Fig. 2](https://anonymous.4open.science/r/ICML-2025-rebuttal-42ED/Fig2.jpg).

---

> > ### Comment · Reviewer_CFgz · 2025-04-04
> >
> > Thanks for the detailed rebuttal. I would like to keep my recommendation for the paper.

---

> > > ### Author Response · Authors · 2025-04-04
> > >
> > > We are sincerely grateful for your careful reading and thoughtful comments,  which have been invaluable in enhancing the clarity and rigor of our work. We hope our responses could answer your questions and doubts.
> > >
> > > Best regards

---

### Decision · Program_Chairs · 2025-05-01

**Decision:**

Accept (poster)

**Comment:**

The rebuttal addressed the concerns from the reviewers and provided comprehensive feedback. It is useful for assessing the paper's contributions. All reviewers recommend acceptance after discussion, and the ACs concur. The final version should include all reviewer comments, suggestions, and additional discussion from the rebuttal.